# Principled Fast and Meta Knowledge Learners for Continual Reinforcement Learning

## Abstract

Inspired by the human learning and memory system, particularly the interplay between the hippocampus and cerebral cortex, this study proposes a dual-learning framework comprising a fast learner and a meta learner to address continual Reinforcement Learning (RL) problems. These two learners are coupled to perform distinct but complementary roles: the fast learner focuses on knowledge transfer, while the meta learner ensures knowledge integration. Unlike traditional multi-task RL approaches that share knowledge via average return maximization, our meta learner incrementally integrates new experiences by explicitly minimizing catastrophic forgetting, and then transfers accumulated knowledge to a single fast learner. To support rapid adaptation to new environments, we introduce an adaptive meta warm-up mechanism that selectively leverages past knowledge. We perform experiments in the pixel-based benchmark and continuous control problems, revealing the comprehensive performance of continual learning for our proposed dual learning approach relative to baseline methods.

## 1 Introduction

Most deep reinforcement learning (RL) algorithms [35, 24, 32, 16, 33] are designed for a single task, where the environment's dynamics and reward function often remain stationary over time. In contrast, humans continuously face diverse and evolving environments, learning to solve new tasks sequentially throughout their lives. Building artificial agents with similar adaptive capabilities requires continual learning — the ability to acquire new knowledge efficiently without forgetting previously learned skills. In this realm, *Continual Reinforcement Learning* [21, 1] emerges as a crucial paradigm, aiming to balance plasticity (rapid adaptation to new tasks) and stability (retention of past knowledge). An ideal continual agent transfers useful knowledge forward to accelerate learning in new environments while avoiding catastrophic forgetting [12] across previously encountered tasks. This fundamental challenge has garnered increasing attention, with broad applications in areas such as Large Language Model (LLM) [43].

Recent work in continual RL spans a range of strategies to address this trade-off [5, 20, 19, 6, 18, 14, 40, 38, 4, 37, 8]. Approaches include synaptic consolidation [18], behavioral cloning across historical policies [38], sparse prompting [40], policy consolidation [19], and policy subspace building [14]. More relevant advances introduce structured learning dynamics: [4] proposes permanent and transient value functions by performing an interplay between fast and slow learning. A simple yet effective baseline method called Reset & Distill (R&D) [3] is specifically proposed to circumvent the negative transfer issue occurring when the new task to learn arrives. [25, 10] addresses the loss of plasticity through the lens of optimization, either by adopting parameter-free online convex optimization or by maintaining the orthogonality of the weight matrix, to enhance fast adaptation to new contexts while avoiding the interference of existing knowledge from past environments. Another direction is

to seek a trade-off between performance and model size [14, 23] by dynamically increasing neural networks to store past knowledge. For instance, [23] uses a growing policy neural network and applies the attention mechanism to integrate the knowledge from the previous policies and the current state to "self-compose" an internal policy. We provide a more detailed discussion of related work in Appendix A. Despite rapid progress across diverse approaches, continual RL still lacks a strong theoretical foundation or principled guidelines for algorithm development. Many existing methods are proposed empirically or heuristically to trade off stability and plasticity, without quantifying explicit objectives to optimize.

To tackle such limitations, our study contributes to new foundations of continual RL, including the definition of the MDP difference to quantify the similarity between different environments, and a quantitative measure of catastrophic forgetting in both value and policy-based RL. Building on these new theoretical foundations and drawing inspiration from neuroscience, we propose a dual-learner paradigm that mirrors neurobiological principles observed in the learning and memory systems of humans [22]. Specifically, we propose to decompose the overall objective in continual RL into two parts: *knowledge transfer* and *knowledge integration*, elucidating their more profound connection to the transfer and multi-task RL problems [31, 29, 26, 27, 36]. In the continual decision-making systems, we maintain two distinct yet complementary components–namely, a fast learner and a meta-learner, which are analogous to the functional roles of the hippocampus (a fast learner) and the neocortex (a meta learner) in the brain.

- **Knowledge Transfer via Fast Learner**: We propose to leverage a fast learner to rapidly acquire knowledge from a new task by adaptively transferring prior knowledge stored in a meta learner. To circumvent the potential negative transfer issue [3, 23], an *adaptive meta warm-up* strategy is developed by either using a direct parameter initialization or adding a behavior cloning regularization in the early training phase. The function of the fast learner in knowledge transfer resembles the hippocampus. By swiftly encoding the new experiences and discriminating the effectiveness of existing knowledge, the hippocampus, guided by the neocortex, specifically functions to quickly assimilate novel scenarios in response to immediate environmental changes or drifts.

- **Knowledge Integration via Meta Learner**: After assimilating the new knowledge by the fast learner, an incremental knowledge integration incorporates the new experiences into the existing knowledge pool stored in the meta learner. Under the new foundation, the knowledge integration is incrementally updated in the principle of *catastrophic forgetting minimization* under specific divergence metrics. After consolidating old and new experiences, the meta learner enhances the adaptive meta warm-up, facilitating the knowledge transfer in the next environment. The knowledge integration process plays a role akin to the cerebral cortex, which gradually integrates, incorporates, and consolidates new knowledge into the existing cognitive structure in the human brain to build a more generalizable, robust, and stable decision-making system.

**Contributions.** The contributions of our study can be succinctly summarized as follows:

1. We propose new foundations of continual RL, including the definition of MDP difference and the measure of catastrophic forgetting, underpinning the algorithmic innovations in the future.

2. We devise a dual-learner system that incorporates distinct yet complementary fast and meta learners to perform knowledge transfer and knowledge integration. The interplay between fast and meta learners mimics the hippocampal-cortical dialogue observed in the brain's memory systems.

3. We provide comprehensive empirical studies to validate the efficiency of our dual-learner system in discrete and continuous action domains, including pixel-based environments and control tasks.

## 2 Problem Setting and New Foundations

**Problem Setting.** We consider a setting with a sequence of $K$ tasks denoted by $k = 1, ..., K$, where each task $k$ is modeled by a Markov Decision Process (MDP) $\mathcal{M}_k = \langle \mathcal{S}_k, \mathcal{A}_k, P_k, R_k, \gamma \rangle$. Here, $\mathcal{S}_k$ and $\mathcal{A}_k$ denote the state and action spaces, respectively; $P_k : \mathcal{S}_k \times \mathcal{A}_k \to \mathcal{P}(\mathcal{S}_k)$ is the transition dynamics; and $R_k : \mathcal{S}_k \times \mathcal{A}_k \to \mathbb{R}$ is the reward function. We define the action-value function $Q^\pi(s, a) = \mathbb{E}_\pi \left[ \sum_{i=0}^\infty \gamma^i R_{t+i+1} \mid S_t = s, A_t = a \right]$ given the state $s$, the action $a$, and a policy $\pi$. Following common practice in continual RL [38, 21, 23, 4], we adopt the following three assumptions: (1) the same state and action spaces, (2) known task transition boundaries, i.e., semi-continual RL [4], and (3) a training budget, including a moderate model size and an allowable computation cost. We aim to seek an optimal policy that can be generalized favorably across the whole sequence of tasks.

**MDP Distance.** The theoretical analysis in continual RL necessitates a quantitative similarity measure between different environments, e.g., MDP. A desirable definition of the MDP distance should fully consider the variation of both reward functions and state transition dynamics between two MDPs. To this end, we use the distance between MDP-determined optimal Q functions $Q_k^*$ and task-specific optimal policies $\pi_k^*$ in the $k$-th MDP environment to define MDP distance in Definition 1.

**Definition 1.** *(MDP Distance)* *For two finite MDPs $MDP_1 = (\mathcal{S}, \mathcal{A}, R_1, P_1, \gamma)$ and $MDP_2 = (\mathcal{S}, \mathcal{A}, R_2, P_2, \gamma)$, we denote their optimal Q functions as $Q_1^*$ and $Q_2^*$ and the optimal policies $\pi_1^*$ and $\pi_2^*$. The Q-value-based or policy-based MDP distances are defined as $d_Q(Q_1^*, Q_2^*)$ and $d_\pi(\pi_1^*, \pi_2^*)$ under certain divergence or distance $d_Q$ and $d_\pi$, respectively.*

For the value-based RL, the optimal Q functions that we utilize to define the MDP difference accommodate the variations of both transition dynamics and reward functions, but often require the same reward scaling or additional normalization for an equal comparison. Although defining MDP distance based on optimal Q functions is intuitive, the mismatch between Q-values from distinct environments may cause additional optimization issues, which we elaborate in Section 3.1.1. The MDP difference based on two optimal policies is more applicable to policy-based RL, such as policy gradient algorithms in the continuous action domain.

**Catastrophic Forgetting.** Our definition of catastrophic forgetting in continual RL is inspired by *distribution drift* and *catastrophic forgetting* quantified in deep learning literature [11], which we briefly recap in Appendix B. Grounded in the definition of MDP difference in Definition 1, we introduce catastrophic forgetting between two MDPs in Definition 2.

**Definition 2.** *(Catastrophic Forgetting across Two Environments)* *Denote $Q_{k-1}, Q_k$ and $\pi_{k-1}, \pi_k$ as Q functions and policies after training RL algorithms across the $(k-1)$-th and $k$-th environments sequentially. Define $\mu_k^{\pi_k}$ and $\mu_k^{Q_k}$ as the state visitation distributions when a policy $\pi_k$ or a greedy policy $\pi_k^Q$ over $Q_k$ (i.e., $\pi_k^Q(\cdot|s) = \arg\max_a Q_k(s,a)$), interacts with the $k$-th environment. The catastrophic forgetting, denoted by CF, is defined as*

$$CF(Q_{k-1}, Q_k) = \sum_{s,a} \mu_{k-1}^{Q_{k-1}}(s)\pi_{k-1}^Q(a|s)d_Q\left(Q_{k-1}(s,a), Q_k(s,a)\right), \tag{1}$$

$$CF(\pi_{k-1}, \pi_k) = \sum_s \mu_{k-1}^{\pi_{k-1}}(s)d_\pi\left(\pi_k(\cdot|s), \pi_{k-1}(\cdot|s)\right). \tag{2}$$

For each $s$ and $a$, the weights $\mu_{k-1}^{Q_{k-1}}(s)\pi_{k-1}^Q(a|s)$ and $\mu_{k-1}^{\pi_{k-1}}(s)$ characterize the importance when measuring the discrepancy between Q functions and policies. Importantly, we use the old policy $\pi_{k-1}$ ($\pi_{k-1}^Q$) instead of the new one $\pi_k$ ($\pi_k^Q$) to evaluate the weights. This strategy is more reasonable as it measures catastrophic forgetting exactly on states and actions that mattered most in the old task. Conversely, if we use $\pi_k$ ($\pi_k^Q$) for the weight evaluation, we might ignore large Q function or policy changes on old-task-critical states and actions that the new policy $\pi_k$ no longer extends visits.

## 3 Principled FAst and MEta Knowledge Continual RL Learners (FAME)

In this section, we apply the proposed FAME framework to value-based and policy-based RL and elaborate on the coupled updating of the fast and meta learners. Our FAME method is illustrated in Figure 1. In principle, the fast learner aims to rapidly learn the new task guided by the meta learner via the proposed adaptive meta warm-up. Meanwhile, the meta learner consolidates the experience from the preceding meta learner and the current fast learner through a knowledge integration process to minimize catastrophic forgetting.

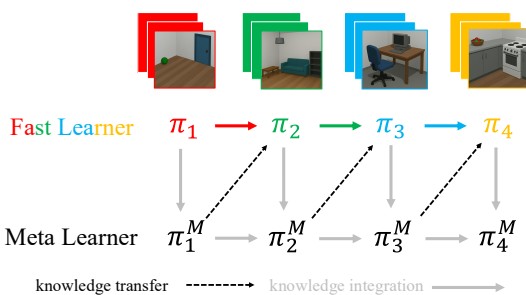

Figure 1: Illustration of FAME. In value-based continual RL, the fast learner can be explicitly denoted by $\{Q_k\}_{k=1}^K$ instead of $\{\pi_k\}_{k=1}^K$.

**Notations.** Let $[N]$ denote $[1, 2, ..., N]$. In value-based RL, we denote $Q_k$ as the updated fast learner after learning task $k$, followed by a meta learner $Q_k^M$ that integrates knowledge from

the preceding meta learner $Q_{k-1}^M$ and $Q_k$. In policy-based RL, we denote $\pi_k$ as the fast learner after learning task $k$, and then a meta learner $\pi_k^M$ integrates knowledge from the preceding meta learner $\pi_{k-1}^M$ and $\pi_k$.

### 3.1 Value-based Continual RL with Discrete Action Space

#### 3.1.1 Knowledge Integration: Catastrophic Forgetting Minimization Principle

After the fast learner $Q_k$ finishes the learning in the $k$-th environment, the FAME approach will transition into a knowledge integration phase, when the meta learner $Q_k^M$ is updated to consolidate information from the past knowledge stored in the preceding meta learner $Q_{k-1}^M$ and the new knowledge acquired by the fast learner $Q_k$. Unlike classical multi-task RL that maximizes the average rewards, our meta learner aims to minimize the catastrophic forgetting defined in Definition 2.

**Q-Value-based Catastrophic Forgetting.** In value-based continual RL, it is natural to first consider the Q-value-based definition of catastrophic forgetting based on Eq. (1). At the $k$-th environment, the optimal meta Q value function $Q_k^M$ is the minimizer by solving the following objective function:

$$Q_k^M = \arg\min_{\widetilde{Q}_k^M} \sum_{i=1}^{k} \sum_{s,a} \mu_i^{Q_i}(s)\pi_i^Q(a|s) \left(Q_i(s,a) - \widetilde{Q}_k^M(s,a)\right)^2, \tag{3}$$

where we recall that $\mu_i^{Q_i}$ is the state visitation distribution when the greedy policy $\pi_i^Q$ (i.e., $\pi_i^Q(\cdot|s) = \arg\max_a Q_i(s,a)$) interacts with the $i$-th environment. Intuitively, the minimizer $Q_k^M$ in Eq. (3) is a weighted average among $\{Q_i\}_{i=1}^k$. However, developing the capability of continual learning by storing all previous Q functions fails to scale in the number of tasks, which is one of the crucial requirements in continual RL. Instead, in Proposition 1 we write the above objective function as an incremental updating rule between the preceding meta learner $Q_{k-1}^M$ and the fast learner $Q_k$. Define the weight function $w_i^Q(s,a) = \mu_i^{Q_i}(s)\pi_i^Q(a|s)$ for each $i \in [k]$. For any measurable function $f(s,a)$ and weight function $w$ with $\sum_{s,a} w(s,a) = 1$ and $w(s,a) \geq 0$ for each $s$ and $a$, we define $\mathbb{E}_w[f] = \sum_{s,a} w(s,a)f(s,a)$. The proof of Proposition 1 is provided in Appendix C.1.

**Proposition 1** (Incremental Q-Value-based Meta Learner Update). *Consider $d_Q$ to be $\ell_2$ loss in Eq.* (1) *in Definition 2. Minimizing Q-value-based catastrophic forgetting in Eq. 3 is equivalent to:*

$$Q_k^M = \arg\min_{\widetilde{Q}_k^M} \sum_{i=1}^{k-1} \mathbb{E}_{w_i^Q}\left[\left(Q_{k-1}^M - \widetilde{Q}_k^M\right)^2\right] + \mathbb{E}_{w_k^Q}\left[\left(Q_k - \widetilde{Q}_k^M\right)^2\right]. \tag{4}$$

**Limitations of Q-Value-based Catastrophic Forgetting.** Proposition 1 leads to an efficient incremental update rule of the meta learner to minimize the principled catastrophic forgetting we define in Definition 2, but it is mainly applicable to distinct environments with similar scales of Q values, such as the ones with varying transition dynamics yet the same reward function. As the new arriving environment is agnostic, the scale of the Q-values may be hard to learn because it is not necessarily bounded and can be quite unstable [29]. The previously well-learned tasks with high rewards tend to be more salient in consolidating knowledge than those with small rewards [44]. Therefore, the policy-based definition of catastrophic forgetting in Eq. (2) is more versatile than the Q-value-based one in Eq. (1), serving as a preferable alternative. In addition, policies may inherently enjoy lower variance than value functions, contributing to improved performance and stability [15].

**Policy-based Catastrophic Forgetting.** Even in value-based continual RL, it is more recommended to employ the policy-based definition of catastrophic forgetting in Eq. (2) to conduct an incremental update of the meta learner. To elaborate, we momentarily go back to the policy-based continual RL setting. At the $k$-th environment, the optimal meta policy $\pi_k^M$ is the minimizer by solving the following objective function:

$$\pi_k^M = \arg\min_{\widetilde{\pi}_k^M} \sum_{i=1}^{k} \sum_{s} \mu_i^{\pi_i}(s)d_\pi\left(\pi_i(\cdot|s), \widetilde{\pi}_k^M(\cdot|s)\right). \tag{5}$$

**Incremental Softmax Meta Learner Update for Value-based Continual RL.** When equipped with the categorical representation, the Q-values can be converted into a Softmax (Boltzmann)

policy, allowing the value-based continual RL to minimize the policy-based catastrophic forgetting objective defined in Eq. (5). Specifically, given a temperature $\tau$, we denote $\pi_i^Q(a|s) = \exp\left(Q_i(a|s)/\tau\right)/\sum_{a'}\exp\left(Q_i(a'|s)/\tau\right)$. By employing the KL divergence as $d_\pi$, in Proposition 2, we replace the meta learner update rule in Proposition 1 by a simpler form.

**Proposition 2** (Incremental Softmax Q-Value-based Meta Learner Update). *Denote* $\widetilde{\pi}_k^M(a|s) = \exp\left(\widetilde{Q}_k^M(a|s)/\tau\right)/\sum_{a'}\exp\left(\widetilde{Q}_k^M(a'|s)/\tau\right)$. *After a softmax policy transformation, the Q-value-based meta learner incremental update is rewritten as*

$$Q_k^M = \arg\min_{\widetilde{Q}_k^M} \sum_{i=1}^{k-1} \mathbb{E}_{w_i^Q}\left[\log\frac{\pi_{k-1}^M}{\widetilde{\pi}_k^M}\right] + \mathbb{E}_{w_k^Q}\left[\log\frac{\pi_k^Q}{\widetilde{\pi}_k^M}\right] = \arg\max_{\widetilde{Q}_k^M} \sum_{i=1}^{k} \mathbb{E}_{w_i^Q}\left[\log\widetilde{\pi}_k^M\right]. \quad (6)$$

The proof of Proposition 2 is straightforward, but we still provide it in Appendix C.2 for completeness. Interestingly, minimizing the policy-based catastrophic forgetting in Eq. (6) is simply solving a Maximum Likelihood Estimator (MLE) by fitting the meta learner $Q_k^M$ to a mixture of state-action distributions across encountered environments. We highlight that this specific objective in Eq. 6 is simplified without relying on $Q_{k-1}^M$ and $Q_k$. However, the knowledge integration in principle consolidates the knowledge from $Q_k$ to $Q_k^M$, resulting in an incremental update rule.

### 3.1.2 Knowledge Transfer via Adaptive Meta Warm-Up

**Challenges.** An effective knowledge transfer necessitates rapidly adapting to the new environment by taking advantage of the previous knowledge if accessible. However, the commonly used finetuning is effective when tasks are similar, but can lead to *negative transfer* issue that frequently occurs in continual RL [3, 38]. The negative transfer, a crucial factor of the *loss of plasticity* [12], leads to performance degradation owing to the dissimilarity between the two tasks. Training from scratch (i.e., reset) is easy to implement to circumvent the negative transfer [9, 3]. However, this naive warm-up lacks flexibility and fails to make full use of the accumulated knowledge to speed up the adaptation to a new task.

**Adaptive Meta Warm-Up via One-vs-all Hypothesis Test.** Alternatively, initializing the fast learner when a new task arrives with the parameters from the meta learner is a straightforward strategy, but previously acquired knowledge and skills may be misleading when learning in a new scenario. For example, it is particularly evident that humans make incorrect decisions or take suboptimal actions when new information contradicts earlier experiences. To harmonize the three conflicting objectives, we propose the adaptive meta warm-up approach to choose the most effective initialization or warm-up strategy among the preceding meta learner, a reset, and the preceding fast learner (i.e., finetune). The adaptive meta warm-up can be framed mathematically within a one-vs-all hypothesis test via policy evaluation in the early interaction phase with a new environment. When the $k$-th environment arrives, we have access to three types of warm-up learners, including the fast learner $Q_{k-1}$, a meta policy $\pi_{k-1}^M$ with the softmax transformation from $Q_{k-1}^M$, and a random Q function $Q^0$ associated with the policy $\pi^0$. The three warm-up learners produce the expected returns, which we define as $V_k^f = \mathbb{E}_{\pi_{k-1}}[R]$, $V_k^M = \mathbb{E}_{\pi_{k-1}^M}[R]$, and $V_k^r = \mathbb{E}_{\pi^0}[R]$. For each task $k$ that arrives, the one-vs-all hypothesis test with a composite null is expressed as

$$H_0 : V_k^M \leq \max\left\{V_k^f, V_k^r\right\} \quad \text{vs.} \quad H_1 : V_k^M > \max\left\{V_k^f, V_k^r\right\}. \quad (7)$$

When the null hypothesis $H_0$ cannot be rejected, we can further compare $V_k^f$ and $V_k^r$ via a common parametric hypothesis test, e.g., t-test. In most scenarios, picking the best warm-up strategy according to the empirical ranking often performs favorably. However, in safety-critical scenarios, e.g., autonomous driving, we must have a rigorous statistical test either by bootstrapping or anytime valid inference [28] on the adaptively collected dataset used for the policy evaluation.

**Meta Warm-Up via Behavior Cloning Regularization.** Once we reject $H_0$, we are ready to perform the meta warm-up. However, directly initializing the fast learner $Q_k$ via the meta policy $\pi_{k-1}^M$ is infeasible as the meta learner is now represented as a policy instead of a Q function under the update in Proposition 2. An easy and effective way to address this policy to value transfer is to impose a Behavior Cloning (BC) regularization in the early training phase, when the meta policy $\pi_k^M$ serves as the expert for data collection and early exploration. Concretely, $Q_k$ is the minimizer of the BC

regularized loss $L(Q_k) = L_0(Q_k) + \lambda\mathbb{E}_s\left[\text{KL}(\pi_{k-1}^M(\cdot|s)||\pi_k^Q(\cdot|s))\right]$, where $L_0(Q_k)$ is the original loss to update $Q_k$, such as the MSE or Huber loss in DQN [24].

### 3.1.3 Algorithm

**Meta Buffer $\mathcal{M}$ in Knowledge Integration.** In the *last* $N$ steps of updating the fast learner in each environment, we additionally store the state-action pairs in a meta learner's buffer $\mathcal{M}$, which are used to approximate $w_i^Q$ for $i \in [k]$ in Eq. (6). Note that the stored state-action pairs are only a small portion of the training dataset for each task (around $1\%$ in our experiments), contributing to a moderate size of the meta buffer $\mathcal{M}$. The moderate size of a meta buffer is crucial, as we are not expected to store too much past data in continual RL. Additionally, we also observe that an overly large $N$ degrades the catastrophic forgetting as the collected state-action pairs in the earlier phase of training are less accurate to approximate $w_i(s, a)$ (see the ablation study in Appendix E.3).

**Algorithm.** We first denote the buffer of the fast learner as $\mathcal{F}$ and $Q^0$ as the randomly initialized Q function. We denote $T$ as the timesteps in each environment. As suggested in Algorithm 1, when the $k$-th environment arrives, we warm start the fast learner $Q_k$ via the adaptive meta warm-up strategy among the preceding meta learner $Q_{k-1}^M$, the preceding fast learner $Q_{k-1}$ and a random learner $Q^0$ (reset) within the first $L$ steps. The adaptive meta warm-up makes full use of previous information to perform an adaptive knowledge transfer. Once the $k$-th task ends,

---

**Algorithm 1** Value-based FAME Update in the $k$-th Environment

1: **Initialize**: Fast Buffer $\mathcal{F}$, Meta Buffer $\mathcal{M}$, $Q_{k-1}^M$, $Q_{k-1}$, $Q^0$, Warm-Up Step $L$, Estimation Step $N$.
2: # Knowledge Transfer: Adaptive Meta Warm-Up
3: Initialize $Q_k$ in $\{Q_{k-1}, Q_k^M, Q^0\}$ via Eq. (7) within $L$ steps
4: **for** $t = L$ to $T$ **do**
5:     Observe $S_t$, take action $A_t$, receive $R_t$, observe $S_{t+1}$
6:     Store $(S_t, A_t, R_t, S_{t+1})$ in $\mathcal{F}$
7:     Update $Q_k$
8:     **if** $t > T - N$ **then**
9:         Store $(S_t, A_t)$ in $\mathcal{M}$ # To Estimate $w_k^Q$
10:     **end if**
11: **end for**
12: Reset $\mathcal{F}$
13: # Knowledge Integration: Minimize Catastrophic Forgetting
14: Update $Q_k^M$ via Eq. (6) using state-action pairs in $\mathcal{M}$

---

the knowledge integration phase starts, when the meta learner $Q_k^M$ is updated via Eq. (6) on the data collected in the meta buffer $\mathcal{M}$. The meta learner $Q_k^M$ incorporates the acquired knowledge in $Q_k$ into $Q_{k-1}^M$ via an incremental update rule in principle.

## 3.2 Policy-based Continual RL with Continuous Action Space

### 3.2.1 Knowledge Integration: Catastrophic Forgetting Minimization Principle

As opposed to the meta learner update with a softmax transformation in Eq. (6) of Proposition 2 for the value-based continual RL, we directly minimize the policy-based catastrophic forgetting in Eq. (5) in terms of the parameterized policy function. The detailed incremental update rule depends on the choice of $d_\pi$ and how we represent the continuous policy in a continuous action space. Next, we will introduce two variants of policy-based continual RL methods when equipped with the forward KL divergence and Wasserstein distance, respectively.

**Method 1 (FAME-KL): Policy Distillation under Forward KL Divergence.** We show that the policy-based knowledge integration will reduce to a policy distillation. Akin to Proposition 2 for value-based continual RL, the policy-based knowledge integration in Eq. (5), when we employ the forward KL divergence and have an accessible probabilistic policy, adopts an update rule of the form:

$$\pi_k^M = \arg\max_{\widetilde{\pi}_k^M} \sum_{i=1}^k \mathbb{E}_{w_i}\left[\log \widetilde{\pi}_k^M\right], \tag{8}$$

where we recall that $w_i(s, a) = \mu_i^{\pi_i}(s)\pi_i(a|s)$ is the policy-based steady state-action distribution on the $i$-th environment. Importantly, this catastrophic forgetting objective above aligns with the knowledge distillation update in policy distillation [29] and typical multi-task RL, such as [36]. Unlike our knowledge integration update, one related continual RL method [23] applies the attention mechanism among all policies to self-compose an internal policy, which is the counterpart of $\pi_k^M$.

**Method 2 (FAME-WD): Wasserstein Distance (WD)-based Knowledge Integration.** Note that when using forward KL divergence, the specific objective of policy-based catastrophic forgetting in Eq. (8) is independent of $\pi_M^{k-1}$ and $\pi_k$. However, the knowledge integration in general should be an incremental update. In Proposition 3, we derive the policy-based incremental update rule under the Wasserstein distance. The proof is given in Appendix C.3.

**Proposition 3** (Incremental Policy-based Meta Learner Update under Wasserstein Distance). *Consider $d_\pi$ to be the squared 2-Wasserstein distance in Eq. (2) of Definition 2. The policy is represented as an independent (multivariate) Gaussian distribution over the action $a$. Minimizing policy-based catastrophic forgetting in Eq. (5) is equivalent to:*

$$\pi_M^k = \arg\min_{\widetilde{\pi}_k^M} \left\{ \sum_{i=1}^{k-1} \sum_s \mu_i^{\pi_i}(s) W_2^2 \left( \widetilde{\pi}_k^M(\cdot|s), \pi_{k-1}^M(\cdot|s) \right) + \sum_s \mu_k^{\pi_k}(s) W_2^2 \left( \widetilde{\pi}_k^M(\cdot|s), \pi_k(\cdot|s) \right) \right\}. \tag{9}$$

In most policy-based algorithms, the policy function is represented by (multivariate) Gaussian distributions, allowing us to easily utilize this incremental update rule to perform the knowledge integration for the meta learner.

### 3.2.2 Knowledge Transfer via Adaptive Meta Warm-Up

For the adaptive meta warm-up in policy gradient methods, we first perform the one-vs-all hypothesis test in Eq. (7) via conducting policy evaluation across $L$ steps, which is proposed in value-based continual RL. Once we determine the best-performing policy, we directly initialize the fast policy in a new task among the fast policy $\pi_{k-1}$, the meta policy $\pi_{k-1}^M$, and a random policy $\pi^0$. Using parameter initialization as the meta warm-up strategy is more convenient for deployment than adding the BC regularization used in Section 3.1.2 in the value-based continual RL.

**Algorithm.** The general description of our policy-based FAME algorithm is similar to Algorithm 1, which is thus provided in Appendix D.

## 4 Experiments

In this section, we validate our FAME approach across a sequence of tasks from multiple environments and domains, including the pixel-based tasks with a discrete action space in Section 4.1 and control problems with a continuous action space in Section 4.2. The central hypothesis is that the interplay between knowledge transfer and knowledge integration of the fast and meta learners in FAME benefits both forward transfer (i.e., plasticity) and catastrophic forgetting (i.e., stability).

**Evaluation Metrics.** We employ the standard metrics [39, 38] in continual RL to evaluate *average performance*, *forgetting* to measure stability, and *forward transfer* to quantify plasticity. Consider $p_i(t)$ to be the success rate or average returns in task $i$ by using the policy at time $t$ with $t \in [K \cdot T]$, where $K$ is the number of environments, and $T$ is the total timesteps in each task. $p_i(t)$ is task-specific with $p_i(t) \in \mathbb{R}$ for our pixel-based tasks and $p_i(t) \in [0, 1]$ in our control tasks.

- **Average Performance.** The average performance is evaluated on the policy at the time $t$ across all $K$ tasks by $P_K(t) = \frac{1}{K} \sum_{i=1}^K p_i(t)$. By default, the average performance is calculated on the final policy when $t = K \times T$. For FAME, this metric is calculated on the meta learner.

- **Forward Transfer (FT)**: The forward transfer is defined as the normalized area between the training curve of the considered algorithm and the baseline. Namely, $FT = \frac{1}{K} \sum_{i=1}^K FTr_i$ with

$$FTr_i = \frac{AUC_i - AUC_i^b}{1 - AUC_i^b}, \quad AUC_i = \frac{1}{\Delta} \int_{(i-1)\cdot\Delta}^{i\cdot\Delta} p_i(t)dt, \quad AUC_i^b = \frac{1}{\Delta} \int_{(i-1)\Delta}^{i\Delta} p_i^b(t)dt. \tag{10}$$

To evaluate this metric in pixel-based tasks, we first normalize $p_i(t)$ in each task to ensure $AUC_i \in [0, 1]$ and then we calculate a normalized metric of the forward transfer.

- **Forgetting (F)**: Forgetting is the performance difference between the policy at the end of a task and after the whole sequence of tasks. Namely, $F = \frac{1}{K} \sum_{i=1}^K F_i$ with $F_i = p_i(i \cdot T) - p_i(K \cdot T)$.

**Experimental Setup.** We perform experiments on the pixel-based tasks from MinAtar environment [41] and robotics arm manipulation tasks from Meta-World [42] with the standard sequence

used in [10]. MinAtar is a standard continual RL benchmark [4] with relatively lighter computational requirements, allowing us to sweep a range of hyperparameters and to report statistical results averaged over 30 seeds. We use breakout, freeway, and spaceinvaders games, and run for 3.5M steps by randomly choosing each of the three games every 500k steps, i.e., 7 tasks in each sequence. DQN [24] is employed to optimize methods in the pixel-based tasks. For manipulation tasks, following the common practice of the literature [23], we deploy the Soft Actor-Critic (SAC) algorithm [16] with $1M$ timesteps on each task and have 10 tasks in each sequence.

### 4.1 Pixel-based Environments with Discrete Action Spaces

**Comparison Methods.** Following [4], we compare our `FAME` approach with DQN (`Reset`), DQN-Finetune (`Finetune`), DQN with a large buffer (`LargeBuffer`), DQN with multi-heads that knows the task identity (`MultiHead`), PT-DQN [4]. Both fast and meta learners in our FAME method employ the same DQN architecture. Except for `Finetune`, we reset the parameters of all baseline methods when each new environment arrives. By contrast, `FAME` applies the adaptive meta warm-up among fast, random initialization, and initial learning with behavior cloning regularization in Section 3.1.2. More details of our experimental setup and hyperparameters are given in Appendix E.1.

**Main Results.** Table 1 summarizes the metric scores of all methods, demonstrating that `FAME` consistently outperforms other baselines in improving knowledge transfer and retaining all knowledge to mitigate catastrophic forgetting. Notably, for the average performance, `FAME` is most stable with minimal variations among all algorithms except for `PT-DQN`, for which the *permanent value function* (i.e., the counterpart of the meta learner) in

Table 1: Main Results on MinAtar on Average Performance (*Avg. Perf*), Forward Transfer (*FT*), and Forgetting. Results (Mean $\pm$ SE) are averaged over 10 sequences, each with 3 seeds. $\uparrow$ denotes a positive metric (more is better), while $\downarrow$ is a negative one (less is better). *Reset* is the baseline for evaluating FT. Forgetting is normalized by the standard deviation in each task.

| Method | Ave. Perf $\uparrow$ | | | FT $\uparrow$ | Forgetting $\downarrow$ |
|---|---|---|---|---|---|
| | Breakout | Spaceinvader | Freeway | | |
| Reset | $6.51 \pm 1.67$ | $3.29 \pm 3.09$ | $0.74 \pm 0.38$ | $0.00 \pm 0.00$ | $1.31 \pm 0.23$ |
| Finetune | $10.62 \pm 2.75$ | $4.95 \pm 2.92$ | $0.89 \pm 0.49$ | $0.13 \pm 0.03$ | $1.26 \pm 0.32$ |
| MultiHead | $6.85 \pm 1.76$ | $3.26 \pm 2.99$ | $0.94 \pm 0.42$ | $-0.01 \pm 0.00$ | $1.25 \pm 0.22$ |
| LargeBuffer | $10.71 \pm 2.84$ | $3.24 \pm 2.91$ | $1.16 \pm 0.59$ | $\mathbf{0.16 \pm 0.02}$ | $1.65 \pm 0.33$ |
| PT-DQN | $0.39 \pm 0.02$ | $0.00 \pm 0.00$ | $0.00 \pm 0.00$ | $0.07 \pm 0.02$ | $1.64 \pm 0.02$ |
| FAME | $\mathbf{14.54 \pm 0.58}$ | $\mathbf{18.72 \pm 0.52}$ | $\mathbf{1.69 \pm 0.17}$ | $\mathbf{0.16 \pm 0.03}$ | $\mathbf{0.72 \pm 0.13}$ |

[4] has limited capability to retain the knowledge and thus keeps almost zero average performance. Regarding the forward transfer, `LargeBuffer` performs similarly to `FAME` as storing more past knowledge also contributes to adapting to a known environment. We also provide the learning curves of all algorithms in Appendix E.2, and an ablation study about $\lambda$, Warm-Up step $L$, and Estimation step $N$ in Appendix E.3.

**Performance of Knowledge Integration.** Figure 2 (left) presents the average performance of all methods at the end of each task, reflecting the tendency of catastrophic forgetting. It turns out that `FAME` achieves the highest average performance in the whole training process in most cases, validating the effectiveness of the meta learner in retaining information through the knowledge integration.

**Performance of Adaptive Meta Warm-Up in Knowledge Transfer.** Figure 2 (right) exhibits the warm-up selection ratio when the agent encounters different types of arriving environments. Concretely, if the agent has already stored relevant data previously in $\mathcal{M}$ about the arriving environment, the meta warm-up is chosen with a $95.1\%$ probability. When a new task occurs against the agent's knowledge, the random initialization is more commonly selected in the adaptive meta warm-up.

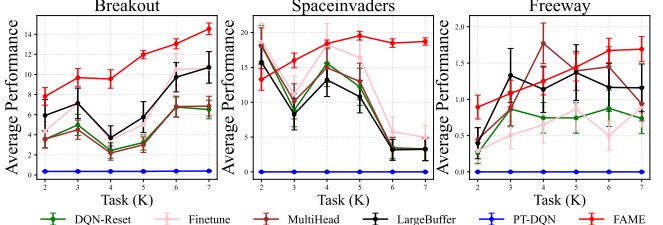
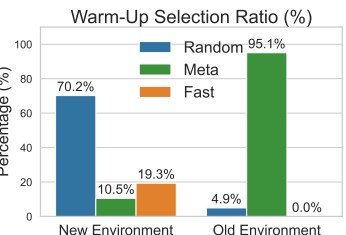

Figure 2: **(Left)** Average performance of the policy at the end of each task across 10 sequences, each of which is averaged over 3 seeds. The vertical lines at each point represent the standard errors. **(Right)** The selection ratio among three warm-up strategies when the arriving environment is old or new.

## 4.2 Robotic Manipulation Tasks with Continuous Action Spaces

**Comparison Methods.** (1) `Reset`; (2) `FineTune`; (3) `Average`: we average the Temporal Difference (TD) targets among all past tasks in evaluating the critic loss; (4) `FAME-KL`: we employ knowledge integration under KL in Eq. (8) (Method 1); (5) `FAME-WD`: we apply knowledge integration under Wasserstein distance in Eq. (9) (Method 2). All methods share the same network architecture as standard SAC. In adaptive meta warm-up, we perform the policy evaluation for 10 episodes among a random policy and the preceding fast and meta policies, and then initialize the fast policy with the best-performing one. The collected data in evaluation is also stored in the fast learner's replay buffer $\mathcal{F}$ without incurring additional interaction costs with the environment. More experimental details of our `FAME` methods (4) and (5) are provided in Appendix F.1.

**Main Results.** As exhibited in Table 2, both `FAME-KL` and `FAME-WD` outperform the baselines significantly across the three metrics. In particular, the superior forward transfer indicates that the adaptive meta warm-up boosts the fast learner's ability to adapt to a new environment by leveraging prior knowledge from the meta learner. Moreover, the highest average performance and minimal forgetting of our `FAME` approaches highlight that the meta learner

Table 2: Main Results on Meta-World on Average Performance (*Ave. Perf*), Forward Transfer (*FT*), and Forgetting. Results are presented as averages and standard errors across 3 seeds.

| Methods | Avg. Perf ↑ | FT ↑ | Forgetting ↓ |
|---|---|---|---|
| Reset | $0.07 \pm 0.05$ | $0.00 \pm 0.00$ | $0.76 \pm 0.08$ |
| Finetune | $0.03 \pm 0.03$ | $-0.36 \pm 0.08$ | $0.39 \pm 0.09$ |
| Average | $0.00 \pm 0.00$ | $-0.56 \pm 0.07$ | $0.10 \pm 0.06$ |
| FAME-WD | $0.87 \pm 0.06$ | $0.04 \pm 0.04$ | $0.03 \pm 0.03$ |
| FAME-KL | $\mathbf{0.93 \pm 0.05}$ | $\mathbf{0.07 \pm 0.04}$ | $\mathbf{0.02 \pm 0.04}$ |

consolidates all past knowledge by conducting an incremental update in knowledge integration.

**Performance of Knowledge Transfer.** To more comprehensively verify the knowledge transfer benefit due to the adaptive meta warm-up in `FAME`, we present the performance profile [2] that reflects the overall performance of the fast learner over the whole sequence of tasks. Figure 3 (left) showcases that both `FAME` methods consistently outperform all baselines, substantiating that the meta learner effectively consolidates knowledge over time and contributes to knowledge transfer. For reference, all learning curves are provided in Appendix F.2.

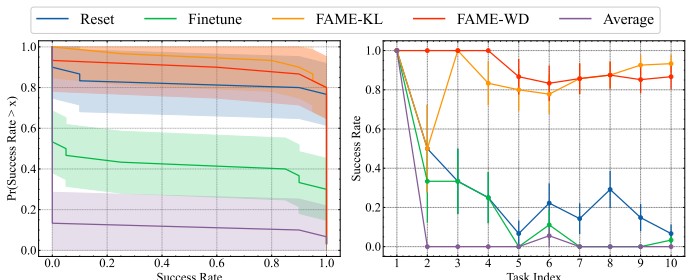

Figure 3: **(Left)** Performance profile of the fast learner across tasks, where the y-axis shows the proportion of tasks that achieve a success rate greater than or equal to the x-axis value. **(Right)** Average performance over time by evaluating the average success rates in the past tasks.

**Performance of Knowledge Integration.** To reflect the tendency of the catastrophic forgetting of `FAME`, we also illustrate the average performance of the meta learner at the end of each task. As suggested in Figure 3 (right), `FAME-KL` and `FAME-MD` enjoy the highest average performance (i.e., minimal catastrophic forgetting) over time across all encountered tasks.

## 5 Discussions and Conclusion

**Limitations and Future Work**. In this study, a meta learner is utilized to retain all knowledge, but it is also possible to conduct an incremental update of the latent representation that can not only distill all knowledge but also perform efficient reasoning to guide the adaptation to a new environment. Beyond the proposed adaptive meta warm-up, more techniques in knowledge transfer can be explored in the future, such as guided exploration and context embedding. Extending our algorithm to the full continual RL context without knowing the task boundary is also valuable for practitioners.

In this paper, we contribute to the foundation of continual RL and develop a novel dual learning system to conduct the knowledge transfer and integration via the coupled update of fast and meta knowledge learners. Two ideas might be worth reemphasizing here. Hypothesis tests or other statistical inference methods are helpful for adaptively selecting practical prior knowledge to overcome the negative transfer issue. Deriving an incremental update rule based on existing multi-task learning objectives is necessary to connect continual and multi-task RL.

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
