# OpenReview forum: "Principled Fast and Meta Knowledge Learners for Continual Reinforcement Learning"
_NeurIPS.cc/2025/Conference — Submitted to NeurIPS 2025_

### Official Review · Reviewer_J9Sz · 2025-07-02

**Clarity:** 2
**Significance:** 2
**Originality:** 2
**Rating:** 4
**Confidence:** 4

**Summary:**

This paper introduces FAME (Fast and Meta knowledge learners), a novel framework for continual reinforcement learning that draws inspiration from the human brain’s memory systems. It proposes a dual-learner architecture comprising a fast learner, which rapidly adapts to new tasks, and a meta learner, which incrementally integrates knowledge to prevent catastrophic forgetting. To support efficient adaptation and stability, the authors define principled metrics such as MDP distance (for task similarity) and catastrophic forgetting (for performance degradation). A key contribution is the adaptive meta warm-up mechanism, which selects the best initialization strategy for new tasks using statistical hypothesis testing, thereby mitigating negative transfer. Theoretical foundations are coupled with practical algorithms for both value-based and policy-based RL, using divergence-based incremental updates. Experiments on MinAtar and Meta-World benchmarks show that FAME achieves superior average performance, forward transfer, and forgetting resistance compared to strong baselines, highlighting its effectiveness in balancing plasticity and stability in continual learning.

**Questions:**

1. The adaptive warm-up mechanism relies on a hypothesis test between fast, meta, and reset strategies using early task returns. However, early rollouts can be noisy or sparse, especially in high-variance environments. How sensitive is the adaptive warm-up mechanism to noise in early evaluations?
2. The current framework assumes known task boundaries, which simplifies the design but limits applicability in non-stationary environments without such information. How well does the method generalize to task-agnostic continual RL?
3. How does the method handle tasks with inconsistent Q-value or reward scales?
4. The framework includes a meta buffer, separate learners, and evaluation steps. While said to be scalable, resource costs are not quantified. Can the authors clarify the memory and computational cost of FAME?

**Ethical Concerns:**

["NO or VERY MINOR ethics concerns only"]

**Final Justification:**

As I explained in my discussion with the authors, I keep my score as it is.

**Limitations:**

The authors have included a clear discussion of the limitations in Section 5, acknowledging issues such as: (1) The assumption of known task boundaries, (2) the scalability limits of the meta learner, (3) and opportunities for extending to task-agnostic continual RL and representation-level knowledge integration. They do not claim that their approach fully solves continual RL, and they appropriately suggest directions for future improvement.

Regarding societal impact, they mark it as “Not Applicable”, which is reasonable in this case. The work is theoretical and methodological in nature, without direct application to sensitive or high-risk domain. However, if deployed in real-world systems, one could imagine downstream impacts where continual learning could cause unexpected behavior due to forgetting or transfer misalignment. This could be briefly noted in future revisions for completeness.

**Paper Formatting Concerns:**

Figure 1 (depicting FAME architecture) is conceptually important but could be visually improved for clarity e.g., more explicit depiction of the interaction between fast and meta learners (maybe add labels on the arrows), and clearer separation between phases.

Some notational choices (e.g., mixing superscripts/subscripts for meta learners vs. task indices) could be cleaned up for readability, though this is not a violation of formatting guidelines. For example, in eq. 9, isn't $\pi^k_M$ supposed to be $\pi^M_k$?

**Quality:**

2

**Strengths And Weaknesses:**

Strengths: (1) The paper demonstrates solid theoretical grounding with clear derivations and well-justified design choices. (2) It introduces quantitative definitions for MDP distance and catastrophic forgetting, advancing the formal treatment of continual RL. (3) Mathematical formulations (e.g., CF definitions, update rules) are intuitive and well-motivated. (4) The principled decomposition into knowledge transfer (via fast learner) and integration (via meta learner) provides a modular foundation for future research and extensions.

Weaknesses: (1) The performance of the Q-value-based update may degrade when Q-values have inconsistent scales across tasks (acknowledged but not deeply addressed in the main paper). (2) No ablation or sensitivity study is presented for the hypothesis test-based warm-up mechanism, which may introduce fragility under noisy returns or limited samples. (3) The exposition could benefit from clearer visual diagrams summarizing the update mechanisms or data flow between the fast and meta learners. (4) No results are shown on longer sequences of tasks (e.g., 10+ tasks), which would better test scalability and forgetting. (5) The neuroscience motivation, while engaging, does not clearly influence the architecture beyond analogy.

---

> ### Author Rebuttal · Authors · 2025-07-31
>
> We thank the reviewer for the insightful feedback and positive assessment of our work, especially on solid theoretical grounding, advancing the formal statement of CRL, intuitive mathematical formulation, and principled learning decomposition. We aim at addressing the concerns you raised in your review.
>
>
> > **Q1: [Weakness (1)]** The performance of the Q-value-based update may degrade when Q-values have inconsistent scales across tasks.
>
> **A1:** This potential performance degradation in the presence of inconsistent scales exactly motivates the policy-based definition of catastrophic forgetting (**Lines 168 to 171**), which we show is more versatile than the Q-value-based one and serves as a preferable alternative. In addition, policies may inherently enjoy lower variance than value functions, contributing to improved performance and stability.
>
> To address this issue, we propose Softmax transformation (**starting from Line 177**) to transform the Q-value-based incremental update to a policy-based incremental update under KL divergence (**in Proposition 2**). This softmax transformation strategy fundamentally solves the inconsistent scale issue across tasks, as aligning Q-functions directly across tasks can be more brittle. In contrast, aligning policies is cleaner in mathematical formulation and more robust against diverse settings.
>
>
>
> > **Q2: [Weakness (2)]** No ablation or sensitivity study is presented for the hypothesis test-based warm-up mechanism, which may introduce fragility under noisy returns or limited samples.
>
> **A2:** Increasing the policy evaluation steps $n$ for the hypothesis test is helpful in general under noisy returns at the cost of reducing the policy optimization updates. In our experiments, we consistently let $n=600$ across all sequences of games on MinAtar as we find it is sufficient to maintain favorable CRL-relevant metrics. In **Table 6 of Appendix E.3**, we have provided the ablation study by varying the policy evaluation steps $n$, suggesting that increasing the policy evaluation step does not significantly impact the Continual RL metrics.
>
> > **Q3: [Weakness (3)]** The exposition could benefit from clearer visual diagrams summarizing the update mechanisms or data flow between the fast and meta learners.
>
> **A3:** Thanks for this great suggestion, and we promise to improve **Figure 1** to better exhibit the update mechanisms and interplay between the fast and meta learners.
>
> > **Q4:[Weakness (4)]** No results are shown on longer sequences of tasks (e.g., 10+tasks), which would better test scalability and forgetting
>
> **A4:** Most existing continual RL benchmarks normally use 10 tasks in the sequence. For example, we follow the MinAtar setting in [5] with 7 games in the sequence and the Meta-world environment in [2,3] with 10 tasks. We will evaluate a longer sequence of tasks in the revised paper and put the results in the appendix for reference.
>
>
> > **Q5:[Weakness (5)]** The neuroscience motivation, while engaging, does not clearly influence the architecture beyond analogy.
>
> **A5:** We believe this is a fundamental challenge to fully connect the growing of neurons in brains with the continual RL algorithm design. Although this research direction is beyond the current scope of our paper, we fully acknowledge this important future direction in continual RL.
>
>
> > **Q6:[Question 1]** The adaptive warm-up mechanism relies on a hypothesis test between fast, meta, and reset strategies using early task returns. However, early rollouts can be noisy or sparse, especially in high-variance environments. How sensitive is the adaptive warm-up mechanism to noise in early evaluations?
>
> **A6:** We employ 600 policy evaluation steps, and it turns out that in our experiments the choice of best warm-up is not sensitive to the potentially noisy returns in the early rollouts. In particular, when the adaptive warm-up mechanism chooses the meta warm-up, we often find the average rewards evaluated by the meta policy are much higher than the other two in our experiments. We agree that there may be some very noisy environments when the trustworthy policy evaluation and comparison become more important, such as the safety-critical scenarios we mentioned in Lines 218-219. This could motivate follow-up work to specifically investigate the policy evaluation effect in noisy environments in the context of RL, but for our experiments, we have not found the sensitivity of the policy's values in the early rollout.
>
> > **Q7:[Question 2]** The current framework assumes known task boundaries, which simplifies the design but limits applicability innon-stationary environments without such information. How well does the method generalize to task-agnostic continual RL?
>
> **A7:** Most continual RL methods assume the task boundary to be known in the literature, as no access to the task boundary will pose a fundamental challenge for continual RL. We think the hypothesis test in the adaptive meta warm-up strategy can be very useful to address this issue. In particular, we can develop an online hypothesis test to discriminate the current data distribution within a sliding time window and the new distribution of the new arriving samples potentially from a new task. This will be a very interesting research direction that relies on the advanced hypothesis test techniques, and we leave it as future work.
>
>
> > **Q8:[Question 3]** How does the method handle tasks with inconsistent Q-value or reward scales?
>
> **A8:** Please refer to **A1**.
>
> > **Q9:[Question 4]** The framework includes a meta buffer, separate learners, and evaluation steps. While said to be scalable,resource costs are not quantified. Can the authors clarify the memory and computational cost of FAME?
>
> **A9:** In terms of space complexity, the dual learner system of FAME requires an additional memory copy as the fast learner (the normal learner in baselines such as Reset and Finetune), and an additional meta replay buffer with the same size as the buffer of the fast size, which is scalable as the total sample memory is fixed and independent of the number of future tasks.
>
> In terms of the computational cost, we employ the same number of agent's interaction steps with the environment for a fair comparison. In other words, the policy evaluation occurs at the cost of reducing the policy optimization update steps in total. The main additional computation cost is the updating of the meta-learner, which is conducted in a supervised learning way. We found it only takes around a couple of minutes across 200 epochs to update the meta-learners in MinAtar after one task finishes. This additional computation overhead is negligible to the overall computation cost in the online training of RL algorithms.
>
> Overall, FAME has a negligible computation cost increase and double space memory to achieve competitive continual RL performance, which is much more efficient and scalable.
>
> > **Q10:Limitations and Paper Formatting**
>
> **A10:** We acknowledge that in Section 5 we discuss the limitations of FAME approach and point out some interesting future directions. We thank the reviewer for mentioning the potential societal impact, and we promise to add this discussion in future revisions.
>
> Regarding the paper formatting, we are grateful to the reviewer for highlighting these issues. We commit to improving the presentation of **Figure 1** for better clarity and cleaning up potential confusion and typos in notations.
>
> Thank you again for pointing out these potential areas of improvement. We appreciate your suggestions. Please let us know if you have any further comments or feedback.
>
> ## Reference
>
> [1] Tianhe Yu, Deirdre Quillen, Zhanpeng He, Ryan Julian, Karol Hausman, Chelsea Finn, Sergey Levine. Meta-world: A benchmark and evaluation for multi-task and meta reinforcement learning. **Conference on robot learning. PMLR, 2020**.
> [2] Malagon, Mikel, Josu Ceberio, and Jose A. Lozano. Self-composing policies for scalable continual reinforcement learning. **ICML 2024**.
> [3] Wesley Chung, Lynn Cherif, David Meger, Doina Precup. Parseval regularization for continual reinforcement learning. **NeurIPS 2024**.
> [4] Hongjoon Ahn, Jinu Hyeon, Youngmin Oh, Bosun Hwang, Taesup Moon. Prevalence of Negative Transfer in Continual Reinforcement Learning: Analyses and a Simple Baseline. **ICLR 2025**.
> [5] Anand, Nishanth, and Doina Precup. Prediction and control in continual reinforcement learning. **NeurIPS 2023**
> [6] Rusu, A. A., Rabinowitz, N. C., Desjardins, G., Soyer, H., Kirkpatrick, J., Kavukcuoglu, K., Pascanu, R., and Hadsell,R. Progressive neural networks. arxiv 2016
> [7] Mallya, A. and Lazebnik, S. Packnet: Adding multiple tasks to a single network by iterative pruning. **CVPR 2018**

---

> > ### Comment · Reviewer_J9Sz · 2025-08-08
> >
> > Thank you for the detailed and thoughtful rebuttal. I appreciate the authors’ clarifications and their clear intention to improve the manuscript in both content and presentation. While some limitations remain, particularly around generalization to more diverse or realistic settings, the paper presents a well-motivated contribution to continual RL. I maintain my original score and recommendation.

---

> > > ### Author Response · Authors · 2025-08-09
> > > **Thank you so much!**
> > >
> > > Thank you very much for your positive evaluation of our work. We sincerely appreciate your time, thoughtful feedback, and kind support throughout the review process.

---

### Official Review · Reviewer_wVDA · 2025-07-03

**Clarity:** 2
**Significance:** 2
**Originality:** 3
**Rating:** 4
**Confidence:** 3

**Summary:**

This paper introduces FAME, a dual-learner framework for continual RL. It has two components: (1) a fast learner that rapidly adapts to each new task initialized by the best network among random network, meta learner and the preceding fast learner (2) a meta learner that incrementally consolidates past and current policies or value functions by minimizing a principled measure of catastrophic forgetting. These two components are instantiated as value based methods in discrete action space and instantiated as policy based methods in continuoous action space. Their experiments show that they outperform baselines on MinAtar and Meta-World tasks, improving forward transfer and greatly reducing forgetting.

**Questions:**

1. Does the replay buffer size increase linearly with the task? An illustration would be helpful to clarify the contribution of the work. If not, how does the meta learner resist the catastrophic forgetting in far old tasks? If so, ideally, continual learning takes constant extra space (or no extra space) to save replay data. I suggest a generative replay buffer, e.g., [1], to reduce the space complexity.
2. The authors choose KL divergence and Wasserstein Distance without an explanation. The main results in the experiments section show that FAME-WD is always worse than FAME-KL, so what is the motivation to bring WD in this work? An explanation would be greatly helpful for readers to understand the technical motivation.

[1] Shin, Hanul, et al. "Continual learning with deep generative replay." Advances in neural information processing systems 30 (2017).

**Ethical Concerns:**

["NO or VERY MINOR ethics concerns only"]

**Final Justification:**

The authors tested the algorithm with additional environments and compared against more baselines, which mitigates the limitation in alogrithm evaluation. I understand that completing all experiments on long task sequences requires time. I look forward to seeing the results on these longer sequences, as mentioned in the response to Reviewer J9Sz, as they could provide strong support for the paper. Additionally, the theoretical justification to Wasserstein distance is clear and convincing and I'm satisfied about it.

**Limitations:**

Yes

**Paper Formatting Concerns:**

In Appendix A, Related Work section, the authors cite papers numbered above 45, but the reference list has only 44 papers and there is no extra reference list in appendix.

**Quality:**

2

**Strengths And Weaknesses:**

Strengths:
 - This paper gives rigorous deifnition of MDP distance and comprehensive mathematic proofs for meta learner updating rools, which gives the work a solid foundation.
 - This paper takes inspiration from biology that proposes a hippocampus-like faster learner and cortex-like meta learner. Such inspiration brings a new perspective to continual learning research.
 - The authors develop theories for both value-based and policy-based algorithms for FAME, making it applicable to broader usecases.

Weakness:
 - The authors perform the one-vs-all hypothesis to initialize a fast learner. Considering that there are three candidate learners including a randomly initialized one, the time and space complexity to perform the hypothesis could be a concern to apply this mechanism in real-world application. It would be nice to have an ablation study how much computational overhead it takes and how much improvements it gains compared to using one of them only
 - The baselines in this paper are too weak to evaluate. There are lots of works in continual RL [1]. I suggest the authors compare with SOTA continual RL algorithms to make the results more convincing.
- Additionally, running experiments on three different tasks repeatedly in shuffled order is not enough to show the knowledge transfer and resistance to catastrophic forgetting of this work. Referring to [1], a typical benchmark runs 6 different Atari games to evaluate a continual RL algorithm. I suggest the authors to run more tasks to make the evaluation more solid.

[1] Powers, Sam, et al. "Cora: Benchmarks, baselines, and metrics as a platform for continual reinforcement learning agents." Conference on Lifelong Learning Agents. PMLR, 2022.

---

> ### Author Rebuttal · Authors · 2025-07-31
>
> We would like to sincerely express our gratitude to the reviewer for their time and effort in reviewing our paper. We appreciate the reviewer's positive recognition of our work on a solid mathematical foundation, a new perspective from biology, and broader application in both value- and policy-based algorithms. We address each concern you raised and are happy to answer any further questions.
>
> > **Q1: [Weakness 1]** The authors perform the one-vs-all hypothesis to initialize a fast learner. Considering that there are threecandidate learners including a randomly initialized one, the time and space complexity to perform the hypothesis could be a concern to apply this mechanism in real-world application. It would be nice to have an ablation study how much computational overhead it takes and how much improvements it gains compared to using one of them only.
>
> **A1:** In adaptive meta warm-up with the one-vs-all hypothesis test, it essentially implies that we replace the policy optimization in the early training phase with the policy evaluation given that we maintain the number of agent interactions with the environment for a fair comparison with other baseline methods. The computation within the policy evaluation steps of our adaptive meta warm-up is even lower than the normal training with the policy optimization throughout, as we do not need to update the policy parameters in the policy evaluation. After the policy evaluation, we conduct the hypothesis test among three scalars with negligible overhead. Overall, the computational overhead of our approach with the adaptive meta-warm-up strategy is also the same as the normal training, e.g., Reset and Finetune method.
>
> Regarding the ablation study among the fast, meta, and random learners, if there is only a random learner, our fast learning only selects the random learner as an initialization, degenerating to a Reset method. If there is only a fast learner accessible, our fast learning degrades to the Finetune method. In conclusion, two baseline methods (Finetune and Reset) are the special case in our adaptive meta warm-up strategy, and the performance of them naturally serves as the ablation study.
>
> > **Q2: [Weakness 2]** The baselines in this paper are too weak to evaluate. There are lots of works in continual RL [1]. I suggest theauthors compare with SOTA continual RL algorithms to make the results more convincing.
>
> **A2:** We add PackNet [6] as the baseline method in Meta-world, which serves as a powerful baseline in multi-task and continual RL settings. The results are provided in **Tables 1** below.  Notably, Packnet requires the task identifiers and the number of all tasks to be known, which is less practical in real applications. Although keeping track of model parameters of the past tasks in Packnet contributes to zero forgetting, but it is less beneficial in the other two metrics compared with our FAME approaches. We also add more baseline methods in **Atari games in Table 2** in **A3**.
>
>
> **Table 1**: Average Results in Meta-world across **three** Sequences over 10 seeds (3 seeds on PackNet). The best and second-best methods in each metric are in bold and underlined, respectively.
>
> | Methods | Avg. Perf | FT |Forgetting|
> |:-:|:-:|:-:|:-:|
> |Reset | 0.093 ± 0.017  | $\\underline{0.000 ± 0.000}$ | 0.710 ± 0.030 |
> |Finetune | 0.037 ± 0.011  | -0.265 ± 0.028 | 0.427 ± 0.033 |
> |Average| 0.013 ± 0.007  | -0.530 ± 0.024 | 0.070 ± 0.022|
> |PackNet| 0.434 ± 0.049 | -0.215 ± 0.020 | **0.000 ± 0.000** |
> |FAME-KL|$\\underline{0.733 ± 0.026}$ | **0.022 ± 0.015** | 0.073 ± 0.019|
> |FAME-WD|**0.767 ± 0.024** | -0.003 ± 0.014 | $\\underline{0.023 ± 0.015}$ |
>
>
> > **Q3: [Weakness 3]** Additionally, running experiments on three different tasks repeatedly in shuffled order is not enough to show the knowledge transfer and resistance to catastrophic forgetting of this work. Referring to [1], a typical benchmarkruns 6 different Atari games to evaluate a continual RL algorithm. I suggest the authors to run more tasks to make the evaluation more solid.
>
> **A3:** We follow the MinAtar benchmark, which is established based on the shuffled order in [5]. Although shuffling order becomes less challenging, it often happens in real life and is particularly useful to demonstrate when our meta warm-up is most suitable when the old environment occurs again. Following your suggestions, we conduct experiments on Atari games by following [2], and add more baseline methods, including PackNet [6] and ProgressiveNet [7]. The result is provided in **Table 2**.
>
> **Table 2**: Results on Atari Games, including ALE/Freeway-v5 environment with 7 different playing modes and ALE/SpaceInvaders-v5 environment with 10 different playing modes. The implementation is adapted from [2]. The best and second-best methods in each metric are in bold and underlined, respectively.
>
>
> | Methods | Freeway: Avg. Perf | Freeway: FT | SpaceInvader: Avg. Perf | SpaceInvader: FT |
> |:-:|:-:|:-:|:-:|:-:|
> |Reset | 0.16 ± 0.18  | 0.00 | 0.10 ± 0.22 | 0.00 |
> |Finetune | 0.21 ± 0.17  | $\\underline{0.53}$ | 0.61 ± 0.41 | **0.65** |
> |ProgressiveNet| 0.39 ± 0.25  | 0.21 | $\\underline{0.61 ± 0.03}$| 0.06 |
> |PackNet| $\\underline{0.41 ± 0.24}$ | 0.18 | 0.47 ± 0.06 | 0.17 |
> |FAME|**0.90 ± 0.12** | **0.68** | **0.96 ± 0.02** | $\\underline{0.63}$ |
>
>
> > **Q4: [Questions 1]** Does the replay buffer size increase linearly with the task? ....
>
>
> **A4:** We allow the number of collected data in the meta buffer to linearly increase with the task **under the constraint of the replay buffer size**. However, we only store a small proportion ($1\\%$ in our experiments) of state and action pairs in the last training phase of each task into the meta buffer (Line 232 in Section 3.1.3), which is very efficient in space complexity. That contributes to a meta buffer with the **same size** as the original buffer in baseline methods, e.g., Reset and Finetune. Our performance is achieved under this constraint for a fair comparison. We can expect the performance improvement if we allow a larger size of replay buffer to take advantage of more historical data.
>
> It may be possible to investigate the generative replay buffer, but training the generative replay buffer may still be vulnerable to the (catastrophic) forgetting issue, i.e., losing the generation ability on the old samples when new samples are continually used to update. Therefore, the efficiency of the samples collected in the replay buffer seems more pressing, and our FAME approach enjoys a high sample efficiency in learning the meta-learner.
>
>
> > **Q5:[Questions 2]**  The authors choose KL divergence and Wasserstein Distance without an explanation. The main results in theexperiments section show that FAME-WD is always worse than FAME-KL, so what is the motivation to bring WD inthis work?
>
> **A5:** The KL and Wasserstein distances are perhaps the most commonly used distances in RL when comparing the difference of policy distribution. The KL divergence is much easier for computation, while it is known that Wasserstein distance is able to capture more data geometry and can be more suitable to compare more complicated policy distributions at the cost of heavy computation. When Gaussian policy is used (which is common in policy-based RL, e,g., PPO, DDPG), the Wasserstein distance can be simplified, allowing us to derive an incremental meta-learner update rule in Proposition 3.
>
> In **Table 2** of our paper, FAME-KL outperforms FAME-WD, but in **Figure 3 (right)**, FAME-WD has higher average performance in the early tasks of the sequence. In addition, in **Table 7 in Appendix F.2** on other sequences of tasks, FAME-WD still performs better in many cases across some metrics. In conclusion, introducing FAME-MD is more theoretically grounded, and we do observe the superiority of FAME-MD in some cases. We will add a detailed explanation in the revised paper to help readers to understand this theoretical motivation and technical contribution.
>
> Thanks again for the reviewer's consistent dedication to reviewing our work, and we are happy to answer any further questions or concerns. Should this rebuttal address your concerns, we would be grateful for an increased score.
>
> ## Reference
>
> [1] Tianhe Yu, Deirdre Quillen, Zhanpeng He, Ryan Julian, Karol Hausman, Chelsea Finn, Sergey Levine. Meta-world: A benchmark and evaluation for multi-task and meta reinforcement learning. **Conference on robot learning. PMLR, 2020**.
> [2] Malagon, Mikel, Josu Ceberio, and Jose A. Lozano. Self-composing policies for scalable continual reinforcement learning. **ICML 2024**.
> [3] Wesley Chung, Lynn Cherif, David Meger, Doina Precup. Parseval regularization for continual reinforcement learning. **NeurIPS 2024**.
> [4] Hongjoon Ahn, Jinu Hyeon, Youngmin Oh, Bosun Hwang, Taesup Moon. Prevalence of Negative Transfer in Continual Reinforcement Learning: Analyses and a Simple Baseline. **ICLR 2025**.
> [5] Anand, Nishanth, and Doina Precup. Prediction and control in continual reinforcement learning. **NeurIPS 2023**
> [6] Rusu, A. A., Rabinowitz, N. C., Desjardins, G., Soyer, H., Kirkpatrick, J., Kavukcuoglu, K., Pascanu, R., and Hadsell,R. Progressive neural networks. arxiv 2016
> [7] Mallya, A. and Lazebnik, S. Packnet: Adding multiple tasks to a single network by iterative pruning. **CVPR 2018**

---

> > ### Author Response · Authors · 2025-08-06
> > **Thank you for review and look forward to the response**
> >
> > Dear Reviewer wVDA,
> >
> > We sincerely value your great suggestions and dedicated guidance in helping us enhance our work. We are eager to ascertain whether our responses adequately address your primary concerns, particularly through our efforts in strengthening the experiments via providing more baselines and a more detailed explanation about the experiments. We would be grateful for the opportunity to provide any needed further feedback. If we have addressed all of your concerns, we kindly ask if you could consider updating your score. Thank you so much!
> >
> > Warm regards,
> > Authors

---

> ### Comment · Reviewer_wVDA · 2025-08-06
>
> Great thanks to the authors for addressing my concerns! The illustrations of computational and space overhead, as well as experiments are comprehensive. I updated my score.
>
> 1. However, I remain concerned about FAME-WD. Although the theoretical advantages of the Wasserstein distance are compelling, they do not consistently translate into empirical improvements over FAME-KL. Is it significantly more difficult to train a FAME-WD model that can outperform FAME-KL? If so, its greater computational cost and mixed results could limit its practical applicability. In the final version, I would appreciate guidance on scenarios in which the Wasserstein distance offers clear benefits, or at least some recommendations for selecting hyperparameters that optimize FAME-WD’s performance.
> 2. As noted in my comments:
> > In Appendix A, Related Work section, the authors cite papers numbered above 45, but the reference list has only 44 papers and there is no extra reference list in appendix.
>
> It appears that some citations are missing. I would appreciate these being added in the final version.

---

> > ### Author Response · Authors · 2025-08-07
> > **Thank you for increasing the score!**
> >
> > We are pleased to have addressed most of the reviewer’s concerns and sincerely appreciate your recognition and support of our work. Below are our explanations for your remaining questions.
> >
> > ### Response to Q1
> >
> > **Computational Efficiency.**  The increased computational overhead of FAME-WD is negligible compared to FAME-KL when using standard Gaussian policy parameterizations common in RL in a continuous action space. As detailed in Appendix C.3, Wasserstein distance is generally advantageous to KL divergence in terms of better capturing data geometry at the cost of higher computational cost. However, most RL algorithms are using **Gaussian policy** by outputting mean vectors (and variance matrices) in the continuous action space, under which Wasserstein distance can be simplified as follows:
> > $$W_2^2(p, q)=\left\|\nu_p-\nu_q\right\|_2^2+\operatorname{tr}\left(\Sigma_p+\Sigma_q-2\left(\Sigma_q^{1 / 2} \Sigma_p \Sigma_q^{1 / 2}\right)^{1 / 2}\right)$$
> > with the two Gaussian distributions denoted by $\mathcal{N}\left(\nu_p, \Sigma_p\right)$ and $\mathcal{N}\left(\nu_q, \Sigma_q\right)$. When the policy is represented as an independent (multivariate) Gaussian distribution across the action $a$, it implies that $\Sigma_p$ and $\Sigma_q$ are diagonal (i.e., variables are independent); then the squared 2‐Wasserstain distance can be further simplified as
> > $$W_2^2(p, q)=\left\|\nu_p-\nu_q\right\|_2^2 + \left\|\sigma_p-\sigma_q\right\|_2^2,$$
> > where $\sigma_p$ and $\sigma_q$ are the diagonal vectors of $\Sigma_p$ and $\Sigma_q$, respectively. **This simplification allows us to compute Wasserstein distance with computational cost comparable to KL divergence, as both rely on the same outputs from the policy network.**
> >
> > **Performance Variability.** We acknowledge that FAME-WD does not consistently outperform FAME-KL across all environments. This may stem from the Gaussian policy assumption, which, while enabling efficient computation, may restrict the Wasserstein distance's capacity to fully exploit geometric advantages in more complex distributions. Thus, the empirical benefits may vary depending on the structure and complexity of the task.
> >
> >
> > **Practical Guidance for FAME-WD.** In complex environments, the task involves high distributional shift and the change of stochasticity, where capturing distribution geometry becomes critical. We believe that is when FAME-WD becomes potentially superior to FAME-KL. In addition, when the online learning budget allows learning full covariance matrices $\Sigma_p$ and $\Sigma_q$ accurately, it recovers more of the expressive power of the Wasserstein metric beyond the diagonal approximation, leading to potential outperformance of FAME-WD. However, when the number of interactions is limited, estimating full covariances may introduce large variance, which could limit the practical utility of FAME-WD relative to FAME-KL.
> >
> > **We will include a more detailed discussion in the final version regarding the computational cost, trade-offs, and practical guidance for when FAME-WD is preferable to FAME-KL.**
> >
> >
> > ### Response to Q2
> >
> > Thank you for pointing out the citation errors. We acknowledge that we added a few citations in Related Work of Appendix A. **We will add them in the updated reference in the final version.**
> >
> >
> > Once again, we sincerely thank you for your support and encouragement of our work, which has given us great confidence. We truly appreciate your warm feedback and extend our deepest respect to you.

---

> > > ### Comment · Reviewer_wVDA · 2025-08-07
> > >
> > > Thank you to the authors for the detailed explanation of FAME-WD, particularly regarding its computational cost and application scenarios. I now have a better understanding of the underlying theory!

---

> > > > ### Author Response · Authors · 2025-08-09
> > > > **Thank you so much!**
> > > >
> > > > We are glad our explanations are useful. We sincerely appreciate your time, thoughtful feedback, and kind support throughout the review process.

---

### Official Review · Reviewer_M6fJ · 2025-07-03

**Clarity:** 3
**Significance:** 3
**Originality:** 3
**Rating:** 4
**Confidence:** 4

**Summary:**

This paper is about proposing a new model inspired by the neuroscience, akin to the theory of complementary learning systems with the role of slow learner being replaced by a meta learner. The role of the meta learner is to perform knowledge integration via minimising catastrophic forgetting across previously encountered tasks. In order to ensure minimal negative transfer, the authors proposed an adaptive meta warm-up strategy, which includes using the fast learner, the meta policy and a random Q function. The authors evaluated their approach using Minatar and Metaworld environments, with baseline models including resetting parameters of the deep neural network, fine-tuning parameters of the deep neural network and a permanent and transient model that also attempts to learn using fast and slow timescales. The proposed model in the paper manages to achieve higher forward transfer and reduced forgetting in continual RL settings when compared with the baseline models.

**Questions:**

1. How is the ideas proposed in definition 1 and definition 2 differs or relate to  bisimulation metrics[1]?

2. What about interference? Are there signs of interference for the tasks studied in the experiments? If so, how does the proposed model cope with limiting the effects of catastrophic interference?


[1] Ferns, Norman, and Doina Precup. "Bisimulation Metrics are Optimal Value Functions." UAI. 2014.

**Ethical Concerns:**

["NO or VERY MINOR ethics concerns only"]

**Final Justification:**

I maintain my score as the set up being evaluated seems to be concerning as pointed out by Reviewer cESJ. The problem is that the tasks being cycled are new and has shorter timeframes, making them less suitable for evaluating the capabilities of the agents to overcome forgetting.

**Limitations:**

Yes

**Quality:**

2

**Strengths And Weaknesses:**

# Strength
1. The paper is mostly clear and well-written. Most of the mathematical propositions were well-defined.
2. The proposed idea in the paper seems to be novel and the results do show that the proposed approach seems to be better at mitigating forgetting.

# Weakness
1. The experiments are done with only 3 random seeds. I would recommend performing with at least 5 random seeds to reduce the variability in the experimental results.
2. $\tilde{Q}^{M}_{k}$ is not defined in Eq 3 and 4.
3. The proposed approach requires the knowledge of task information and task boundary. How will the approach fare where task boundary  is unclear? While to the credits of the authors, they did indicate this as a limitation in the limitation section.
4. In line 43-44, the authors mentioned that many existing methods have been proposed that trade off stability and plasticity, without quantifying explicit objectives to optimise. It will be nice if the authors can provide some citations and elaborate what could be some of these explicit objectives to consider.
5. The Minatar benchmark is somewhat simplistic in terms of representation and game dynamics complexity. How would the model fare in more complex games like Atari? I recommend trying 3-5 Atari games, but with less than 50Million steps (200 Million Frames).

---

> ### Author Rebuttal · Authors · 2025-07-31
>
> We thank the reviewer for the positive feedback of our work, especially for the well-established mathematical foundations, the novelty of our approach, and the significant improvement in experiments. We aim at addressing the concerns you raised in your review.
>
>
> > **Q1:[Weakness 1]** The experiments are done with only 3 random seeds. I would recommend performing with at least 5 random seeds to reduce the variability in the experimental results.
>
>
> **A1:** Thank you for this advice. For Meta-world, we now provide an updated summary of results **over 10 seeds** in **Table 1** below, consistently substantiating the superiority of FAME. For the MinAtar environment, we clarify that our experiments are averaged over 10 sequences, each sequence with 3 seeds (30 seeds in total).
>
> **Table 1**: Average Results in Meta-world across **three** sequences over 10 seeds (3 seeds on PackNet). The best and second-best methods in each metric are in bold and underlined, respectively.
>
> | Methods | Avg. Perf | FT |Forgetting|
> |:-:|:-:|:-:|:-:|
> |Reset | 0.093 ± 0.017  | $\\underline{0.000 ± 0.000}$ | 0.710 ± 0.030 |
> |Finetune | 0.037 ± 0.011  | -0.265 ± 0.028 | 0.427 ± 0.033 |
> |Average| 0.013 ± 0.007  | -0.530 ± 0.024 | 0.070 ± 0.022|
> |PackNet| 0.434 ± 0.049 | -0.215 ± 0.020 | **0.000 ± 0.000** |
> |FAME-KL|$\\underline{0.733 ± 0.026}$ | **0.022 ± 0.015** | 0.073 ± 0.019|
> |FAME-WD|**0.767 ± 0.024** | -0.003 ± 0.014 | $\\underline{0.023 ± 0.015}$ |
>
>
> > **Q2:[Weakness 2]** $\tilde{Q}_k^M$ is not defined in Eq 3 and 4.
>
> **A2:** We thank the reviewer for the helpful comment. $\tilde{Q}_k^M$ denotes a candidate Q function used to approximate $Q_k^M$ and we will state this in the revised paper to avoid ambiguity.
>
> > **Q3:[Weakness 3]** The proposed approach requires the knowledge of task information and task boundary. How will the approachfare where task boundary is unclear? While to the credits of the authors, they did indicate this as a limitation in the limitation section.
>
> **A3:** Most continual RL methods assume the task boundary to be known in the literature, as no access to the task boundary will pose a fundamental challenge for continual RL. We think the hypothesis test in the adaptive meta warm-up strategy has great potential to address this issue within the FAME approach. In particular, we can develop an online hypothesis test to discriminate between the current data distribution within a sliding time window and the new distribution of the new arriving samples. This will be a very interesting research direction by leveraging the advanced hypothesis test techniques, and we leave it as future work.
>
> > **Q4:[Weakness 4]** In line 43-44, the authors mentioned that many existing methods have been proposed that trade off stability and plasticity, without quantifying explicit objectives to optimise. It will be nice if the authors can provide some citations and elaborate what could be some of these explicit objectives to consider.
>
> **A4:** For example, [5] decomposes the value function into two with different time scales and optimizes them, but it is not clear whether the permanent value function holds the general knowledge and is able to explicitly reduce catastrophic forgetting. After a careful literature review, we strongly believe the foundations of continual RL, such as a principled measure of MDP distance or similarity (Definition 1), and catastrophic forgetting (Definition 2), play a pivotal role in developing continual RL algorithms in the community. We promise to add more citations to support this claim and add more details of our explicit objective, i.e., definitions 1 and 2, in the revised version of our paper.
>
>
> > **Q5:[Weakness 5]** The Minatar benchmark is somewhat simplistic in terms of representation and game dynamics complexity. How would the model fare in more complex games like Atari?
>
> **A5:** Thanks for this suggestion. The choice of Minatar follows [5]. Albeit with a relatively lighter computational cost, MinAtar allows us to sweep a range of hyperparameters and to analyze the mechanism of our proposed method in detail, such as the selection ratio in Figure 2 (right). To strengthen our empirical evidence, we have also added experiments in Atari games by following [2] and adding more baseline methods. The results are provided in **Table 2** below.
>
>
> **Table 2**: Results on Atari Games, including ALE/Freeway-v5 environment with 7 different playing modes and ALE/SpaceInvaders-v5 environment with 10 different playing modes. The implementation is adapted from [2]. The best and second-best methods in each metric are in bold and underlined, respectively.
>
>
> | Methods | Freeway: Avg. Perf | Freeway: FT | SpaceInvader: Avg. Perf | SpaceInvader: FT |
> |:-:|:-:|:-:|:-:|:-:|
> |Reset | 0.16 ± 0.18  | 0.00 | 0.10 ± 0.22 | 0.00 |
> |Finetune | 0.21 ± 0.17  | $\\underline{0.53}$ | 0.61 ± 0.41 | **0.65** |
> |ProgressiveNet| 0.39 ± 0.25  | 0.21 | $\\underline{0.61 ± 0.03}$| 0.06 |
> |PackNet| $\\underline{0.41 ± 0.24}$ | 0.18 | 0.47 ± 0.06 | 0.17 |
> |FAME|**0.90 ± 0.12** | **0.68** | **0.96 ± 0.02** | $\\underline{0.63}$ |
>
>
> > **Q6:[Question 1]** How is the ideas proposed in definition 1 and definition 2 differs or relate to bisimulation metrics[1]?
>
> **A6:** Thanks for sharing this reference. Albeit being complimentary, bisimulation metrics [1] provide additional evidence to support the optimal value functions that can be naturally used to measure the state similarity, which is consistent with Definition 1. From this perspective, Definition 1 targets the overall state similarity by averaging all states (and actions) to define MDP distance and also extends the MDP difference from the difference of optimal value functions to policies. Based on Definition 1, Definition 2 is considered as a weighted MDP difference with the weights determined by the old tasks and new policy or value functions, serving as the foundation of quantifying catastrophic forgetting.
>
> > **Q7:[Question 1]** What about interference? Are there signs of interference for the tasks studied in the experiments? If so, how doesthe proposed model cope with limiting the effects of catastrophic interference?
>
> **A7:** If we understand correctly, catastrophic interference in continual RL is the loss of performance on previous tasks when learning new ones, which also refers to catastrophic forgetting. The purpose of a meta-learner is exactly to minimize this catastrophic forgetting, defined based on Definitions 1 and 2. Feel free to clarify the questions if you find we have a misunderstanding here.
>
>
> Thank you again for pointing out these potential areas of improvement. We appreciate your suggestions. Please let us know if you have any further comments or feedback.
>
> ## Reference
>
> [1] Tianhe Yu, Deirdre Quillen, Zhanpeng He, Ryan Julian, Karol Hausman, Chelsea Finn, Sergey Levine. Meta-world: A benchmark and evaluation for multi-task and meta reinforcement learning. **Conference on robot learning. PMLR, 2020**.
> [2] Malagon, Mikel, Josu Ceberio, and Jose A. Lozano. Self-composing policies for scalable continual reinforcement learning. **ICML 2024**.
> [3] Wesley Chung, Lynn Cherif, David Meger, Doina Precup. Parseval regularization for continual reinforcement learning. **NeurIPS 2024**.
> [4] Hongjoon Ahn, Jinu Hyeon, Youngmin Oh, Bosun Hwang, Taesup Moon. Prevalence of Negative Transfer in Continual Reinforcement Learning: Analyses and a Simple Baseline. **ICLR 2025**.
> [5] Anand, Nishanth, and Doina Precup. Prediction and control in continual reinforcement learning. **NeurIPS 2023**
> [6] Rusu, A. A., Rabinowitz, N. C., Desjardins, G., Soyer, H., Kirkpatrick, J., Kavukcuoglu, K., Pascanu, R., and Hadsell,R. Progressive neural networks. arxiv 2016
> [7] Mallya, A. and Lazebnik, S. Packnet: Adding multiple tasks to a single network by iterative pruning. **CVPR 2018**

---

> ### Comment · Reviewer_M6fJ · 2025-08-04
>
> I thank the authors for the clarification.
>
> Having read the comments from the authors on A7, I am concerned that the authors have misunderstood the difference between catastrophic interference and catastrophic forgetting. Catastrophic interference refers to the inability of the agent to learn new knowledge after prior learning of other task(s) while catastrophic forgetting refers to the inability of the agent to recall previously learned knowledge.
>
> Can the authors discuss how their approach will fare in the catastrophic interference scenario?

---

> > ### Author Response · Authors · 2025-08-04
> >
> > Thank you very much for the helpful clarification regarding the difference between catastrophic interference and catastrophic forgetting. Based on the definition of catastrophic interference as the inability of the agent to learn new knowledge after prior learning of other task(s), it corresponds closely to the **loss of plasticity**, which is effectively captured by the **Forward Transfer (FT)** metric as the existing literature does. Specifically, a higher FT indicates that the new task learning is less impaired by the prior task learning and thus reflects lower catastrophic interference. Therefore, our metrics in the paper are comprehensive to evaluate the catastrophic interference as well. We appreciate the reviewer raising this point and will incorporate this connection into the final version of the paper.

---

### Official Review · Reviewer_cESJ · 2025-07-03

**Clarity:** 4
**Significance:** 3
**Originality:** 3
**Rating:** 4
**Confidence:** 4

**Summary:**

The paper introduces FAME, a CRL framework that combines a fast learner for task-specific adaptation with a meta-learner that consolidates knowledge to prevent forgetting. Forgetting is defined as a weighted divergence in behavior or value estimates based on prior task visitation. The meta-learner is updated incrementally using a compact buffer of high-quality transitions. To avoid negative transfer, the method employs an adaptive warm-up that selects between meta-initialization, fine-tuning, or reset based on early performance via a hypothesis test. When the meta-policy is used, behavior cloning regularizes early training. Experiments on MinAtar and Meta-World show improved transfer and reduced forgetting over simple baselines.

**Questions:**

1. Why not evaluate FAME on established benchmarks for continual reinforcement learning?
2. Why is FAME not compared against popular or SOTA CL methods?
3. How sensitive is FAME to the choice of temperature in the softmax transformation and the size of the meta buffer?

**Ethical Concerns:**

["NO or VERY MINOR ethics concerns only"]

**Final Justification:**

FAME is a strong contribution to the CRL community; measuring forgetting via MDP differences and using a meta-policy for selective initialization and avoiding negative transfer are both novel and valuable approaches. However, the issue of limited baseline coverage remains after the rebuttal. While adding PackNet improves the evaluation, the absence of more recent and diverse CRL baselines makes it difficult to fully position FAME’s performance in the broader context. I remain positive about the paper, but believe the baseline comparison could be strengthened.

**Limitations:**

Yes

**Paper Formatting Concerns:**

No concerns

**Quality:**

3

**Strengths And Weaknesses:**

# Strengths

1. When confronted with a new task in the sequence, the authors propose to select the best warm-up strategy among three options after running short rollouts with each. At the expense of a small overhead at the beginning of each task, FAME can better avoid the negative transfer problem. The most novel strategy is the use of the meta-policy. By estimating task similarity via either policy or Q-value divergence, the method avoids reusing prior knowledge when it's likely to be harmful. If optimal behaviors differ significantly across tasks, reusing past policies can lead to poor performance. Similarly, if the same state-action pairs yield very different Q-values, it signals a mismatch in task dynamics.
2. The authors introduce a precise, weighted definition of forgetting based on MDP differences, framing it not merely as a drop in performance, as is common in prior work, but as a principled measure of how much the agent's behavior or value estimates diverge from what it previously knew, specifically on the states and actions that mattered most in past tasks. This grounds forgetting in the agent’s own experience, offering a more behaviorally meaningful and task-relevant signal than aggregate performance metrics alone.
3. To update the meta-learner in the value-based setting, the authors apply a clever softmax transformation to convert Q-functions into probability distributions. This allows them to operate in policy space, where forgetting is easier to interpret, optimize (via KL divergence), and align with actual agent behavior, which ultimately matters more than raw Q-values. Aligning Q-functions directly across tasks can be more brittle. In contrast, aligning policies is cleaner and more robust.
4. The memory footprint is low. Only ~1% of the most recent task’s training data is stored in the meta buffer, making it well-suited for continual RL where storage is constrained. Importantly, FAME doesn’t store arbitrary rollouts, but captures samples from the final training phase, when the policy is more likely to be competent. This focused distillation ensures that the meta-learner is updated using higher-quality, task-relevant behaviors.
5. FAME is grounded in strong theoretical foundations, with key components formally derived and supported by proofs. The paper is also clearly written and well-cited, situating the work effectively within the broader literature.

# Weaknesses

1. The evaluation is weak. It is based on seven simplified Atari-style environments and a single 10-task Meta-World sequence, with all results averaged over just three seeds. The authors do not evaluate FAME on established continual RL benchmarks such as Continual World [1], COOM [2], or CoRA [3], despite the fact that these benchmarks align well with FAME's assumptions (shared state/action spaces, known task boundaries, and replayable transitions). This makes it difficult to assess the scalability and generality of the method. In terms of baselines, the paper compares only against simple adaptations of standard RL algorithms, while omitting well-established continual learning methods like EWC, AGEM, MAS, and PackNet. The CRL benchmarks listed above already include many baseline evaluations to directly compare against. For instance, the authors discuss R&D as a motivation, but never include it in the empirical comparisons. As a result, the evaluation does not provide a clear picture of how FAME performs relative to state-of-the-art methods targeting similar objectives.
2. FAME’s adaptive warm-up mechanism appears to provide the most benefit when tasks are revisited, rather than when encountering entirely new environments. As shown in Figure 2 (Right), the warm-up strategy selects the meta-policy 95% of the time when the incoming task has been seen before, but only 19% for genuinely novel tasks. This suggests that much of the advantage stems from re-encountering previously learned tasks. In their MinAtar setup, the authors construct a 7-task sequence using only 4 environments, meaning that task repeats are explicitly included. Revisiting tasks is a less challenging setting than tackling a continual stream of novel tasks in non-repeating, task-incremental scenarios.

[1] Wołczyk, Maciej, et al. "Continual world: A robotic benchmark for continual reinforcement learning." *Advances in Neural Information Processing Systems* 34 (2021): 28496-28510.

[2] Tomilin, Tristan, et al. "Coom: A game benchmark for continual reinforcement learning." *Advances in Neural Information Processing Systems* 36 (2023): 67794-67832.

[3] Powers, Sam, et al. "Cora: Benchmarks, baselines, and metrics as a platform for continual reinforcement learning agents." *Conference on Lifelong Learning Agents*. PMLR, 2022.

---

> ### Author Rebuttal · Authors · 2025-07-31
>
> We thank the reviewer for the appreciation of our work, especially for the novel use of the meta policy, a principled measure of forgetting, the Q-function to policy alignment, and memory efficiency. We aim at addressing the concerns you raised in your review.
>
> > **Q1:** The evaluation is weak. It is based on seven simplified Atari-style environments and a single 10-task Meta-World sequence, with all results averaged over just three seeds.
>
> **A1:** For the MinAtar environment, we clarify that our experiments are averaged over 10 sequences, each sequence with 3 seeds (30 seeds in total). For Meta-world, in the submitted paper, we also provide similar results of two extra sequences of tasks in **Appendix F.2** across 3 seeds, and sequences of tasks are also given. To reduce the randomness, we now provide an updated summary of results for each sequence of tasks **over 10 seeds** (3 seeds on PackNet [7]) in **Tables 1.1-1.3**, consistently substantiating the superiority of FAME-KL and FAME-MD. More explanation of Packnet [7] is given in **A3**.
>
> **Table 1.1**: Results of **Sequence 1** in Meta-world averaged over 10 seeds. The best and second-best methods in each metric are in bold and underlined, respectively.
>
> | Methods | Avg. Perf | FT | Forgetting |
> |:-:|:-:|:-:|:-:|
> |Reset | 0.090 ± 0.029 | 0.000 ± 0.000|0.800 ± 0.040|
> |Finetune | 0.070 ± 0.026 | -0.294 ± 0.039|0.480 ± 0.050|
> |Average|0.020 ± 0.014|-0.584 ± 0.035|0.100 ± 0.030|
> |PackNet|0.620 ± 0.104|-0.115 ± 0.032|**0.000 ± 0.000**|
> |FAME-KL|$\\underline{0.860 ± 0.035}$|**0.042 ± 0.019**|$\\underline{0.050 ± 0.026}$|
> |FAME-WD|**0.870 ± 0.034**|$\underline{0.004 ± 0.022}$|0.010 ± 0.017|
>
> **Table 1.2**: Results of **Sequence 2** in Meta-world averaged over 10 seeds. The best and second-best methods in each metric are in bold and underlined, respectively.
>
> | Methods | Avg. Perf | FT | Forgetting |
> |:-:|:-:|:-:|:-:|
> |Reset |0.110 ± 0.031 | 0.000 ± 0.000|0.680 ± 0.047|
> |Finetune | 0.040 ± 0.020|-0.252 ± 0.045|0.440 ± 0.052|
> |Average|0.000 ± 0.000|-0.496 ± 0.039|0.110 ± 0.031|
> |PackNet|0.393 ± 0.078|-0.304 ± 0.034|**0.000 ± 0.000**|
> |FAME-KL|$\\underline{0.680 ± 0.047}$|$\\underline{0.004 ± 0.029}$|0.120 ± 0.036|
> |FAME-WD|**0.750 ± 0.044**|**0.008 ± 0.024**|$\underline{0.040 ± 0.032}$|
>
> **Table 1.3**: Results of **Sequence 3** in Meta-world averaged over 10 seeds. The best and second-best methods in each metric are in bold and underlined, respectively.
>
> | Methods | Avg. Perf | FT |Forgetting|
> |:-:|:-:|:-:|:-:|
> |Reset | 0.080 ± 0.027|$\\underline{0.000 ± 0.000}$|0.650 ± 0.052|
> |Finetune | 0.000 ± 0.000|-0.250 ± 0.036|0.360 ± 0.048|
> |Average|0.020 ± 0.014|-0.509 ± 0.036|$\\underline{0.000 ± 0.020}$|
> |PackNet|0.350 ± 0.077|-0.226 ± 0.039|**0.000 ± 0.000**|
> |FAME-KL|$\\underline{0.660 ± 0.048}$|**0.022 ± 0.028**|0.050 ± 0.030|
> |FAME-WD|**0.680 ± 0.047**|-0.020 ± 0.028|0.020 ± 0.032|
>
>
>
> > **Q2:** The authors do not evaluate FAME on established continual RL benchmarks such as Continual World [1], COOM [2], or CoRA [3].
>
> **A2:** The Continual World Benchmark is built on top of Meta-World as a testbed, an established benchmark in meta-learning and multi-task learning and continual RL communities. Continual world defines a specific sequence of tasks (e.g., CW10 or CW20) using tasks from Meta-world, while Meta-world offers more flexible evaluation on various sequences of tasks. For example, many recent continual RL work, such as [2,3,4], is evaluating their methods on Meta-world directly. Section 4.2 of our paper is based on Meta-world, with the implementation adapted from [3] (Wesley Chung et al. NeurIPS 2024). Section 4.1 in MinAtar follows [5] (Anand Nishanth et al. NeurIPS 2023) as well as the implementation to better elucidate the mechanism of our FAME approach, especially the adaptive meta warm-up strategy via Figure 2. Our evaluation is highly established on common continual RL benchmarks in existing literature.
>
> > **Q3:** In terms of baselines, the paper compares only against simple adaptations of standard RL algorithms, while omitting well-established continual learning methods like EWC, AGEM, MAS, and PackNet.
>
> **A3:** We add the Packnet in our empirical comparison, as Packnet is considered as a more powerful and widely used baseline as opposed to others. The results are provided in **Tables 1.1-1.3**, where our FAME approach also outperforms Packnet across average performance and forward transfer. Notably, Packnet achieves zero forgetting by storing (masks of) model parameters of previous tasks by additionally knowing the task identifiers and the number of tasks in advance, which is less practical in real scenarios. The average result across three sequences in Meta-world and additional comparisons in Atari games with more baseline methods (ProgressiveNet [6]) are provided in **Tables 1 and 2** below.
>
> **Table 1**: Average Results in Meta-world across **three** Sequences over 10 Seeds (3 seeds on PackNet). The best and second-best methods in each metric are in bold and underlined, respectively.
>
> | Methods | Avg. Perf | FT |Forgetting|
> |:-:|:-:|:-:|:-:|
> |Reset | 0.093 ± 0.017  | $\\underline{0.000 ± 0.000}$ | 0.710 ± 0.030 |
> |Finetune | 0.037 ± 0.011  | -0.265 ± 0.028 | 0.427 ± 0.033 |
> |Average| 0.013 ± 0.007  | -0.530 ± 0.024 | 0.070 ± 0.022|
> |PackNet| 0.434 ± 0.049 | -0.215 ± 0.020 | **0.000 ± 0.000** |
> |FAME-KL|$\\underline{0.733 ± 0.026}$ | **0.022 ± 0.015** | 0.073 ± 0.019|
> |FAME-WD|**0.767 ± 0.024** | -0.003 ± 0.014 | $\\underline{0.023 ± 0.015}$ |
>
> **Table 2**: Results on Atari Games, including ALE/Freeway-v5 environment with 7 different playing modes and ALE/SpaceInvaders-v5 environment with 10 different playing modes. The implementation is adapted from [2]. The best and second-best method in each metric are in bold and underlined, respectively.
>
> | Methods | Freeway: Avg. Perf | Freeway: FT | SpaceInvader: Avg. Perf | SpaceInvader: FT |
> |:-:|:-:|:-:|:-:|:-:|
> |Reset | 0.16 ± 0.18  | 0.00 | 0.10 ± 0.22 | 0.00 |
> |Finetune | 0.21 ± 0.17  | $\underline{0.53}$ | 0.61 ± 0.41 | **0.65** |
> |ProgressiveNet| 0.39 ± 0.25  | 0.21 | $\underline{0.61 ± 0.03}$| 0.06 |
> |PackNet| $\underline{0.41 ± 0.24}$ | 0.18 | 0.47 ± 0.06 | 0.17 |
> |FAME|**0.90 ± 0.12** | **0.68** | **0.96 ± 0.02** | $\underline{0.63}$ |
>
> > **Q4:** FAME’s adaptive warm-up mechanism appears to provide the most benefit when tasks are revisited, rather than when encountering entirely new environments. ... Revisiting tasks is a less challenging setting than tackling a continual stream of novel tasks in non-repeating, task-incremental scenarios.
>
> **A4:** The shuffled order of tasks is particularly useful to illuminate the mechanism of our adaptive meta warm-up strategy. In real applications, the new arriving task is agnostic, which can be either totally unknown or re-encountered. Although a shuffled order of sequence is slightly less challenging to totally non-repeating scenarios, evaluating mixed types of new tasks is also meaningful for general continual RL scenarios (Section 4.1).
>
> The adaptive meta warm-up offers more flexibility to choose the best initialization or warm-up for an agnostic task. For instance, it is very intuitive to choose a meta warm-up more often when an old task is revisited, while it chooses a reset (random initialization) when a new environment occurs. In addition to the scenarios with mixed types of new arriving environments, in Section 4.2, we focus on the non-repeating and task-incremental scenarios, suggesting consistent performance. The continual RL scenarios with shuffled order of tasks and incremental tasks in Sections 4.1 and 4.2 of our experiments provide a more comprehensive evaluation of the continual RL performance.
>
>
>
> > **Q5:** How sensitive is FAME to the choice of temperature in the softmax transformation and the size of the meta buffer?
>
> **A5:** Our FAME method is robust to the choice of hyperparameters for the considered ranges in our ablation study analysis in **Appendix E.3**. For instance, we directly leverage categorical transformation, i.e., Softmax transformation with the temperature $\tau=1$, and keep the maximum size of the meta buffer as the same as the standard on used in DQN-Reset and DQN-Finetune. Our FAME approach under these simple choices of hyper-parameters already suggests desirable performance. The sensitivity analysis of all other hyperparameters with a line search is provided in **Appendix E.3**, including the regularization hyperparameter $\lambda$, the warm-up step $L$, the policy evaluation step $n$, and the weight estimation step $N$, demonstrating that our FAME approach is not sensitive to these hyperparameters for the considered ranges.
>
>
> Thank you again for pointing out these potential areas of improvement. We appreciate your suggestions. Please let us know if you have any further comments or feedback.
>
> ## Reference
>
> [1] Tianhe Yu, Deirdre Quillen, Zhanpeng He, Ryan Julian, Karol Hausman, Chelsea Finn, Sergey Levine. Meta-world: A benchmark and evaluation for multi-task and meta reinforcement learning. **Conference on robot learning. PMLR, 2020**.
> [2] Malagon, Mikel, Josu Ceberio, and Jose A. Lozano. Self-composing policies for scalable continual reinforcement learning. **ICML 2024**.
> [3] Wesley Chung, Lynn Cherif, David Meger, Doina Precup. Parseval regularization for continual reinforcement learning. **NeurIPS 2024**.
> [4] Hongjoon Ahn, Jinu Hyeon, Youngmin Oh, Bosun Hwang, Taesup Moon. Prevalence of Negative Transfer in Continual Reinforcement Learning: Analyses and a Simple Baseline. **ICLR 2025**.
> [5] Anand, Nishanth, and Doina Precup. Prediction and control in continual reinforcement learning. **NeurIPS 2023**
> [6] Rusu, A. A., Rabinowitz, N. C., Desjardins, G., Soyer, H., Kirkpatrick, J., Kavukcuoglu, K., Pascanu, R., and Hadsell,R. Progressive neural networks. arxiv 2016
> [7] Mallya, A. and Lazebnik, S. Packnet: Adding multiple tasks to a single network by iterative pruning. **CVPR 2018**

---

> > ### Comment · Reviewer_cESJ · 2025-08-03
> >
> > I appreciate the authors’ efforts in adding PackNet, running more seeds, and including limited Atari experiments. These changes marginally improve the credibility of the evaluation but do not address my main concerns (also raised by reviewer wVDA).
> >
> > The core issues of **evaluation design and baseline coverage** remain. The authors claim that
> >
> > > "Our evaluation is highly established on common continual RL benchmarks in existing literature."
> >
> > I would argue calling those *highly established*, but rather **custom, self-constructed task sequences** within Meta-World and MinAtar, used in a few prior works. A key advantage of using standardized benchmarks is the **availability of baseline results**, which would address the lack of SOTA comparisons and remove the need for arbitrary task design and ad-hoc reimplementations. Leveraging these benchmarks would have enabled direct, fair, and more convincing evaluations.
> >
> > The rebuttal further claims that
> > > Meta-World offers more flexible evaluation on various sequences of tasks
> >
> > yet offers no explanation of why this flexibility is valuable or preferable. For example, PackNet achieves 0.8 average performance on CW20 (from Continual World), but far less (0.35, 0.39, 0.62) on the authors’ 3 custom sequences. Why is this the case? Claiming that
> > > “recent works also evaluate on Meta-World directly”
> >
> > does not really justify much either.
> >
> > Finally, MinAtar experiments with **repeating tasks** are weak evidence for continual learning. I don’t see the motivation for this in realistic scenarios. An agent is generally expected to learn a task and retain knowledge of it, rather than regaining access to re-learning it.
> >
> > Given these unresolved issues, I am **keeping my score**.

---

> > > ### Author Response · Authors · 2025-08-05
> > >
> > > We express our gratitude to the reviewer for their consistent dedication to reviewing our work. Of course, we remain at your disposal for any further clarifications or questions. Below are our response to your concerns one by one.
> > >
> > > > I would argue calling those highly established, but rather custom, self-constructed task sequences within Meta-World and MinAtar, used in a few prior works.....
> > >
> > >
> > > Thanks for this insightful comment. We acknowledge the value of standardized continual RL benchmarks, e.g., continual world, which provide a fixed sequence of tasks and available baseline results for a more direct and reproducible comparison. However, we would like to also emphasize that our experiments are based on released code of more recently peer-reviewed works, such as [2] (ICML 2024 oral) for Atari games and Meta-world, and [5] (NeurIPS 2023) for MinAtar. These work also **adopt fixed sequences of tasks and available baseline results to allow a fair comparison**. While the sequences of tasks are slightly different from the one in continual world (e.g.,CW10), we believe evaluation on these more recent work and diverse sequences also provide sufficient evaluations, without suffering from the potential overfitting or **selection bias** on a single sequence of tasks in CW10 and CW20 in continual world.
> > >
> > >
> > > Importantly, continual RL remains a relatively open field in terms of agreed-upon benchmarks, and relying solely on one standard sequence of tasks in continual world may overlook the **generality and robustness** of algorithms across varied and realistic scenarios. This is also reflectd in recent works [2,3,4], which adopt broader choices of sequence of tasks in Meta-world beyond the fixed Continualworld sequence, highlighing the community's ongoing interest in evaluating across more diverse and realisti settings.
> > >
> > > Overall, we fully recognize the value of using standardized benchmark with a standardized sequence of task. At the same time, we promise to add the empirical comparison on the standardized sequence of task, i.e., CW10 from contiual world, in the revised version.
> > >
> > > > yet offers no explanation of why this flexibility is valuable or preferable. For example, PackNet achieves 0.8 average performance on CW20 (from Continual World), but far less (0.35, 0.39, 0.62) on the authors’ 3 c...
> > >
> > > Akin to the response above, the flexibility in evaluation means that we are able to evaluate the metrics of continual RL algorithms on **more sequences of tasks** based on Meta-world, instead of one particular sequence of tasks chosen in CW10 in continual world, which likely suffers from the selection bias on the sequence of tasks in the long run. Our implementation of Packnet is directly adapted from [2]. The experiments are conducted on three sequences of tasks following [2], which is different from the one used in Table 1 of continual world.
> > >
> > > By comparing the results of Packnet on the four sequences of tasks, we can find the average performance is still relatively sensitive to the choice of task sequence. This variation in average performance across different sequences also stresses the importance of evaluation on multiple task sequences.
> > >
> > > > does not really justify much either.
> > >
> > > Apologize for the lack of detailed explanation. Many recent continual RL work, such as [2,3,4], is evaluating their methods on Meta-world directly. For example, the experiments in [2] are evaluated on a sequence of meta-world benchmark, and two Atai games with different play modes. In [3], the authors do the evaluation on gridworld, CARL and Meta World tasks. [4] performed experiments on Meta world, Deepmind Control, and Atari-100k. Note that all of these recent work all include Metaworld as the benchmark.
> > >
> > >
> > > > Finally, MinAtar experiments with repeating tasks are weak evidence for continual learning. I don’t see the motivation for this in realistic scenarios. An agent is generally expected to learn a task and retain knowledge of it, rather than regaining access to re-learning it.
> > >
> > >
> > > Even though repeating tasks are less challenging than fully new tasks, we argue that evaluation on repeating tasks is still important in continual RL. In real applications, tasks are **not strictly one-pass**; instead, agents may encounter previously seen tasks or environments with similar structure—especially in dynamic, cyclical, or seasonal settings. For instance, the house robot is required to clean the floor again in the following weeks after they finish it this week.
> > >
> > > Importantly, evaluation on re-encountering tasks is **a crucial supplement** to our non-repeating task experiments in meta-world, contributing to fully investigating the catastrophic forgetting in **truly agnostic environments** in the real continual learning. That's also why CW20, which repeats CW10 twice to evaluate on the same tasks in another round, is also used in addition to CW10 in the continual world.

---

> > > > ### Comment · Reviewer_cESJ · 2025-08-08
> > > >
> > > > Thank you for the extensive clarification regarding using Meta-World tasks to form different sequences than the fixed ones (CW10) present in Continual World. I acknowledge that this is a valid approach.
> > > >
> > > > > agents may encounter previously seen tasks or environments with similar structure—especially in dynamic, cyclical, or seasonal settings.
> > > >
> > > >
> > > >
> > > > I believe there might be some conflation here between evaluating performance on learned tasks and retraining on learned tasks as a means of evaluation. Of course, evaluating learned tasks is essential for measuring retention. In the example you bring, the house robot should indeed be able to clean the floor again, but it should not have to *re-learn* how to clean the floor beforehand. Note also my response to Reviewer M6fJ regarding this matter.
> > > >
> > > > An important part of my motivation for bringing up other CRL benchmarks was to enable stronger baseline comparisons. In the current setup, the only recent baseline is PT-DQN, while the others are largely naive DQN variants. Adding PackNet was a step in the right direction, but it still remains difficult to position FAME’s performance within the broader landscape. That said, I consider FAME a strong contribution; measuring forgetting via MDP differences and using a meta-policy for selective initialization are very interesting approaches.
> > > >
> > > > I remain positive about the paper, but the baseline comparison could be strengthened, so I will keep my current score.

---

> > > > > ### Author Response · Authors · 2025-08-09
> > > > >
> > > > > Thank you very much for your thoughtful follow-up. We greatly appreciate your recognition of the contributions and your suggestions for further improving the experimental evaluation. We agree that incorporating more baselines makes the evaluation stronger. In the final version, we will include more recent CRL baselines beyond PT-DQN and PackNet in MinAtar to strengthen comparative analysis. Thank you again for your valuable feedback and support.

---

> ### Comment · Reviewer_M6fJ · 2025-08-05
>
> Hi Reviewer cESJ,
>
> I hope you don't mind me jumping in here. I would like to seek your clarification on the dismissal of the last point. Are you against the use of Minatar for studying continual RL or the use of repeating tasks for studying continual RL?
>
> It is common in the continual RL literature to use repeating tasks, especially in Atari [8] and Minatar [5]. The latter which the authors have cited. I listed just two papers here but there is much more in the literature.
>
> [5] Anand, Nishanth, and Doina Precup. Prediction and control in continual reinforcement learning. NeurIPS 2023
>
> [8] Abbas, Zaheer, et al. "Loss of plasticity in continual deep reinforcement learning." Conference on lifelong learning agents. PMLR, 2023.

---

> > ### Comment · Reviewer_cESJ · 2025-08-08
> >
> > Thanks for the comment. I see there was ambiguity in my previous response. To clarify, I am not against the use of MinAtar/Atari, nor against repeated sequences for studying continual RL, both are indeed common. The key question is **what is being evaluated**. For measuring *plasticity* (as in [8]), repeated tasks are entirely reasonable. However, when also evaluating *retention*/*forgetting* and *transfer* (as in [5]), they are less meaningful than a setting with only new tasks, especially when the task sequence is as short as in the authors’ work. I don’t fully disregard this approach, but relying solely on short sequences with repeated tasks weakens the evaluation of the authors’ method.

---

### Note · Authors · 2025-08-13

Dear Reviewers and AC,

We sincerely thank all reviewers for their time, effort, and constructive feedback. We are deeply encouraged that multiple reviewers suggest positive assessments of our work from the perspectives of the theoretical foundations in continual RL, the measure of forgetting via MDP difference, the novelty of the proposed meta warm-up mechanism, and the overall improvements over baseline methods.

In response to the reviewers' comments regarding evaluation settings, we have expanded our experiments to include additional task sequences and standardized continual RL benchmarks such as Atari and Meta-World, as well as stronger baselines including PackNet and ProgressiveNet. We have also increased the number of seeds (up to 10) and provided additional ablation and sensitivity analyses to strengthen the empirical evaluations.

We have clarified theoretical aspects (e.g., softmax transformation for policy-based updates) and further explained the motivation, cost, and benefits of using both KL and Wasserstein metrics. We appreciate the feedback regarding limitations, formatting, and broader impact and will incorporate these suggestions into the final version.

In summary, we are glad to receive the positive assessment from the multiple reviewers after the rebuttal phase. We appreciate the thoughtful reviews and will release our code to support reproducibility. We hope the revisions and our responses address the main concerns and that the paper can contribute meaningfully to the community—especially to develop more principled continual RL algorithms in the future.

Sincerely,
Authors

---

### Decision · Program_Chairs · 2025-09-17

**Decision:**

Reject

**Comment:**

This is a borderline paper. Though the reviewers identified a few contributions of this work, they still have a few remaining concerns after rebuttal. 1) A weak evaluation that relies on small, custom setups and excludes established continual RL benchmarks (like Continual World) and leans on task repeats, which are less challenging than novel tasks. 2) Insufficient baseline coverage: the paper only compares to simple RL adaptations and fails to clearly position FAME against state-of-the-art methods.